# REINFORCED LATENT REASONING FOR LLM-BASED RECOMMENDATION

**Yang Zhang**[1]* **Wenxin Xu**[2]* **Xiaoyan Zhao**[1]† **Wenjie Wang**[2] **Fuli Feng**[2]†
**Xiangnan He**[3] **Tat-Seng Chua**[1]
[1]National University of Singapore [2]University of Science and Technology of China
[3]National Engineering Laboratory for BITA, University of Science and Technology of China
zyang1580@gmail.com, xuwenxin@mail.ustc.edu.cn

## ABSTRACT

Large Language Models (LLMs) have demonstrated impressive reasoning capabilities in complex problem-solving tasks, sparking growing interest in their application to preference reasoning in recommendation systems. Existing methods typically rely on fine-tuning with explicit chain-of-thought (CoT) data. However, these methods face significant practical limitations due to (1) the difficulty of obtaining high-quality CoT data in recommendation and (2) the high inference latency caused by generating CoT reasoning. In this work, we explore an alternative approach that shifts from explicit CoT reasoning to compact, information-dense latent reasoning. This approach eliminates the need for explicit CoT generation and improves inference efficiency, as few latent tokens can effectively capture the entire reasoning process. Building on this idea, we propose *Reinforced Latent Reasoning for Recommendation* (LatentR$^3$), a novel end-to-end training framework that leverages reinforcement learning (RL) to optimize latent reasoning without relying on any CoT data. LatentR$^3$ adopts a two-stage training strategy: first, supervised fine-tuning to initialize the latent reasoning module, followed by pure RL training to encourage exploration through a rule-based reward design. Our RL implementation is based on a modified GRPO algorithm, which reduces computational overhead during training and introduces continuous reward signals for more efficient learning. Extensive experiments demonstrate that LatentR$^3$ enables effective latent reasoning without any direct supervision of the reasoning process, significantly improving performance when integrated with different LLM-based recommendation methods. Our codes are available at https://github.com/xuwenxinedu/R3.

## 1 INTRODUCTION

Enhancing the reasoning capabilities of Large Language Models (LLMs) has been a central research objective since the LLMs' emergence, with recent advances—such as DeepSeek-R1 (Guo et al., 2025) and OpenAI-o1 (OpenAI, 2024)—fueling a surge of interest in this direction. By being trained or architected to reason more deliberately through techniques like chain-of-thought (CoT) (Wei et al., 2022) reasoning, LLMs have demonstrated remarkable progress in tackling complex real-world problems (Xu et al., 2025; Shao et al., 2024), rivaling or even surpassing human PhD-level performance in certain cases (OpenAI, 2024). These developments have generated growing enthusiasm for applying LLM reasoning to downstream tasks, including one key AI application for information access—recommender systems (Tsai et al., 2024; Bismay et al., 2025). For recommendation, the core lies in reasoning user preferences from historical behaviors (Tsai et al., 2024), which only implicitly indicate preference. This is well aligned with the strengths of LLM reasoning, offering the potential to unlock new paradigms for personalized recommendation.

Existing approaches to LLM reasoning for recommendation typically rely on explicit textual reasoning (*i.e.*, chain-of-thought, CoT) data to fine-tune models and enhance their reasoning capabilities (Bismay et al., 2025; Tsai et al., 2024; Fang et al., 2025), following trends in general LLM reasoning

---

*Equal contribution.
†Corresponding authors. Email: {xy.zhao2333,fulifeng93}@gmail.com

tasks. However, applying this paradigm in recommendation presents fundamental challenges. First, these methods necessitate generating explicit CoT reasoning during inference, which would incur prohibitive computational costs and latency—critical concerns for real-world deployment. Second, collecting high-quality supervision CoT data for effective tuning is difficult: 1) user feedback in recommendation is usually limited to final outcomes, with no access to underlying reasoning; and 2) the subjective, personalized nature of preferences (Tsai et al., 2024) makes manual annotation or synthesis to obtain CoT data both costly and unreliable. All these limitations restrict the practical applicability of current explicit COT methods. To address this, we pose a central question: *Can we eliminate the need for explicit CoT reasoning at both the tuning and inference, while still unleashing the reasoning potential of LLM for recommendation*?

A promising direction is to move from natural language reasoning to latent reasoning (Hao et al., 2024), where LLMs reason directly in their hidden representation space, eliminating the need for explicit textual CoT reasoning. Moreover, because hidden states have much higher information density than textual tokens, compact latent representations can encode complex reasoning processes. This alleviates the need for lengthy reasoning chains (Su et al., 2025) and thus enables more efficient inference. Yet, existing latent reasoning methods in general domains are not directly applicable to our goal, as they still rely on explicit CoT supervision to learn latent reasoning (Hao et al., 2024), such as by distilling CoT reasoning into latent reasoning.

This work explores learning latent reasoning without relying on any explicit CoT data, in an end-to-end optimization manner. Achieving this is challenging, as the only available signal for reasoning supervision is weak, coming solely from final user feedback, with no direct guidance on the reasoning process itself. Inspired by the recent success of reinforcement learning (RL) in learning explicit CoT reasoning strategies without CoT supervision, such as DeepSeek-R1-Zero (Guo et al., 2025), we investigate the use of RL to achieve this goal. However, directly applying RL training[1] can be unstable and prone to collapse, particularly given the vast, high-dimensional space of latent reasoning. To mitigate this, we adopt a two-stage training strategy inspired by DeepSeek-R1 (Guo et al., 2025). In the first stage, we apply supervised fine-tuning to warm up the latent reasoning module, providing a strong initialization. In the second stage, we conduct pure RL training with a rule-based reward to promote exploration and further improve the reasoning ability.

Taking a step further, our RL approach builds on the GRPO algorithm (Shao et al., 2024), with task-specific modifications to the reward design. In its original main form, GRPO assigns a discrete reward (*e.g.,* binary) by comparing each generated answer to ground truth, requiring full autoregressive answer generation for each sampled reasoning path—resulting in substantial computational overhead. To address this, we modify the reward to use the perplexity of the target item as a proxy, thereby eliminating the need for costly answer generation during training. This also produces a continuous reward signal, which could provide richer learning feedback and may facilitate more efficient optimization. Furthermore, we shift from a group-relative advantage to a batch-relative advantage by using the batch average reward as the baseline for advantage computation. This adjustment addresses the issue in our continuous reward setting, where the group-relative method could assign unreliable positive advantages, even if the entire group exhibits low-quality reasoning. Since our method operates with RL, we refer to it as *Reinforced Latent Reasoning for Recommendation* (**LatentR**[3]).

The main contribution of this work can be summarized as follows:

- We highlight that latent reasoning provides a practical and latency-efficient method for integrating LLM reasoning into recommendation systems and propose to achieve latent reasoning in recommendations without relying on explicit CoT data.

- We propose LatentR, a new method that enables latent reasoning with zero-shot CoT examples through a carefully designed RL framework. It features a novel reward formulation and an improved advantage computation mechanism to guide the optimization process of latent reasoning.

- We conduct extensive experiments on real-world datasets, demonstrating that without explicit CoT data, our latent reasoning approach can yield significant performance improvements.

---

[1]That means performing RL training from scratch.

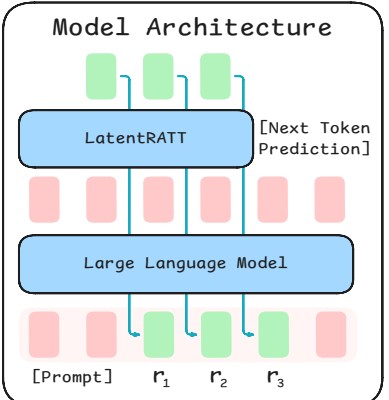 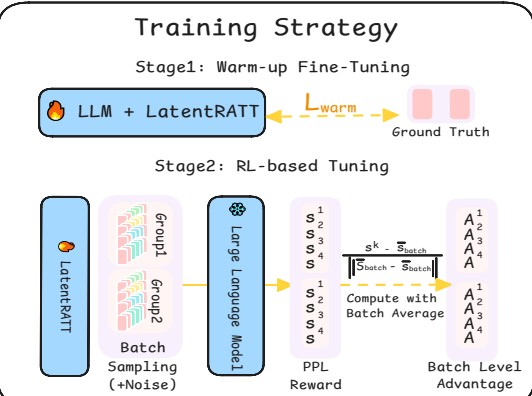

Figure 1: An illustration of the model architecture and training strategy of the proposed LatentR$^3$. On the architectural side, a LatentRATT layer is introduced to generate latent reasoning continuous tokens. The training follows a two-stage framework: the first stage performs warm-up tuning via supervised fine-tuning (SFT), while the second stage applies RL based on a modified GRPO, termed LR-GRPO, to further optimize the reasoning.

## 2 PROBLEM DEFINITION

Let $\mathbb{D}$ denote the collected recommendation data, and let $(u, h, y) \in \mathbb{D}$ denote an instance, where $u$ is a user, $h$ is the user's historical interactions, and $y$ is the next item the user interacts with. Both $h$ and $y$ are described using textual information (e.g., item titles). To leverage the capabilities of LLMs, we reformulate the recommendation task as a natural language problem. Specifically, for each data point $(u, h, y)$, we convert the historical interactions into a textual prompt $x$, which is then input to the LLM to generate a next-item recommendation, denoting the token length as $|x|$. Thereafter, we can also represent a data point by $(x, y)$. Given that user preferences are implicitly and intricately embedded in the historical data, we avoid asking the LLM to produce the recommendation directly. Instead, we introduce a **thinking** process that encourages the model to first perform intermediate reasoning before generating the final output, which can be formulated as:

$$x \xrightarrow{LLM(x)} r \xrightarrow{LLM(x,r)} \hat{y}, \tag{1}$$

where $r$ denotes the LLM's intermediate reasoning output, *i.e.,* thoughts, and $\hat{y}$ is the final predicted item. Importantly, we aim for this reasoning process to be efficient and learnable without requiring additional supervision in the form of explicit reasoning data (CoT annotations). In other words, we train the LLM to perform recommendation-oriented reasoning directly from $\mathcal{D}$, without relying on externally annotated CoT data[2].

## 3 METHODOLOGY

We propose achieving inference-efficient, explicit CoT-free reasoning through reasoning in the latent space and leveraging RL to enable reasoning learning. This forms our method, Reinforced Latent Reasoning for Recommendation (LatentR$^3$). As shown in Figure 1, LatentR$^3$ has key designs in both the reasoning architecture and the learning strategy side:

- **Latent Reasoning Architecture**. To enable latent reasoning, LatentR$^3$ adds an attention layer on top of the LLM's final decoding layer, denoted by "LatentRATT". This layer extracts information from the final hidden states of the LLM to generate latent reasoning tokens (hidden states) aligned with the LLM's input embedding space, in an autoregressive manner. Together, the LLM and this generation layer constitute the full reasoning model.

---

[2]Although Rec-R1 (Lin et al., 2025a) is similar to DeepSeek-R1-zero (learning without CoT), it focuses on learning query rewriting and summarization for recommendation rather than general reasoning.

- **Reinforced Learning.** Given the lack of direct supervision for the reasoning process, the model must learn latent reasoning from final output signals. To enable the model to effectively learn latent reasoning, we use a reinforcement learning approach consisting of two stages: 1) Warm-up Fine-Tuning, which warms up the latent reasoning model through supervised fine-tuning, providing a meaningful initialization for the subsequent RL stage, and 2) RL-based Tuning, which applies pure RL to encourage exploration of reasoning and further enhance its reasoning capabilities. The RL stage is implemented using the GRPO method, with customized designs for sampling, reward, and advantage estimation to align with our specific requirements.

Next, we present the details of our latent reasoning architecture, followed by an explanation of the reinforced learning method.

## 3.1 LATENT REASONING ARCHITECTURE

Before generating the final answer, we guide the LLM to first produce reasoning tokens in a continuous latent space to simulate slow thinking. Unlike prior work that directly uses hidden states from the LLM's decoding layers as latent thoughts within the input embedding space, we introduce an additional attention layer on top of the final decoding layer to explicitly generate latent thought tokens. This layer serves two key purposes: (1) functioning as a specialized reasoning token generator that aggregates contextual information to produce coherent latent thoughts, and (2) aligning the latent reasoning tokens more seamlessly with the LLM's input embedding space.

Formally, given a data point converted into the prompt format $(x, y)$, at the $i$-th reasoning token generation, we first obtain the hidden states from the LLM's final decoding layer by inputting $x$ along with all previously generated latent reasoning tokens. We then apply our additional attention layer to these hidden states and take the last-position output as the next latent reasoning token. Formally,

$$\boldsymbol{H}_{i-1} = LLM^{-1}(x, r_1, ..., r_{i-1}), \tag{2}$$

$$r_i = \text{LatentRATT}(\boldsymbol{H}_{i-1})[-1], \tag{3}$$

where $LLM^{-1}(\cdot)$ denotes the output of the LLM's final decoding layer; $r_1, \ldots, r_{i-1}$ are the previously generated latent reasoning tokens; and $\boldsymbol{H}_{i-1} \in \mathbb{R}^{(|x|+i-1)\times d}$ represents the corresponding sequence of hidden states produced by $LLM^{-1}(\cdot)$, with dimensionality $d$. "LatentRATT" refers to our additional attention layer for generating latent reasoning tokens. Finally, $r_i \in \mathbb{R}^d$ denotes the $i$-th generated latent reasoning token.

*Final Generation.* After generating $N$ latent reasoning tokens to form the final reasoning thought $r = [r_1, \ldots, r_N]$, we concatenate $x$ and $r$ as the input to the LLM for next-item prediction. Here, $N$ is a hyperparameter controlling the length of the latent reasoning sequence, and under our framework, a minimal $N$ (e.g., $N = 1$) can achieve strong performance.

## 3.2 REINFORCED LEARNING

We leverage RL to facilitate the effective learning of latent reasoning. Inspired by DeepSeek-R1, instead of performing RL from scratch, we first use supervised fine-tuning to warm up the model, providing a strong initial model for RL tuning. We then apply the pure RL to explore better reasoning, forming a two-stage tuning strategy. Next, we introduce each of these stages in detail.

### 3.2.1 WARM-UP FINE-TUNING

Training RL from scratch can lead to instability and potential collapse, especially considering the vast, high-dimensional space of latent reasoning. To mitigate this, we first perform SFT to warm up the reasoning model, providing a solid initialization. Specifically, we fine-tune the full model using a standard next-token prediction task, enabling the model to generate meaningful latent reasoning tokens that enhance recommendation performance. The tuning objective is formulated as follows:

$$\min_{\Omega} \quad L_{warm} = -\sum_{(x,y)\in\mathbb{D}} \sum_{i=1}^{|y|} log P_\theta(y_i|x, r, y_{<i}), \tag{4}$$

where $y_i$ denotes the $i$-th token of $y$, $|y|$ denotes the total number of tokens in $y$, and $P_\theta(y_i|x, r, y_{<i}) =$ LLM$(x, r, y_{<i})$ denotes the predicted probability of $y_i$ given the previous generations $y_{<i}$, latent reasoning $r$, and $x$; $\theta$ represents the total model parameters including those of the original LLM and our "LatentRATT" layer, and $L_{warm}$ denotes the final total loss.

### 3.2.2 RL-BASED TUNING

Although the model may have gained some preliminary latent reasoning capabilities through SFT, its performance remains insufficient, as it might focus solely on data fitting rather than exploring optimal reasoning paths, leading to suboptimal outcomes. To address this, we consider leveraging RL training to facilitate exploration of more diverse reasoning paths, using the SFT results from the first stage as the initialization for further learning.

Given the demonstrated success of GRPO in DeepSeek-R1 (Guo et al., 2025), we adopt it as the foundation for implementing our RL training. Generally, GRPO operates in four steps: (1) sampling a set of textual outputs from the old policy model; (2) computing a binary reward for each output; (3) calculating the advantage by comparing the group average reward; and (4) performing policy updates based on the advantage. To better align with our latent reasoning approach and enhance training efficiency, we introduce specific modifications to the first three steps, resulting in our new GRPO variant, termed **LR-GRPO**. In short, (1) we sample latent reasoning tokens in continuous space via the reparameterization trick rather than in textual space, (2) employ perplexity-based rewards, and (3) compute advantages relative to the batch-level average reward instead of the group-level to enhance learning under the new reward design. We now elaborate on these modifications:

• **Sampling.** Since our latent reasoning operates in a continuous space, it cannot be directly sampled like discrete textual tokens in standard GRPO. To address this, we leverage the Reparameterization Trick (Kingma et al., 2013; 2015), treating the generated latent reasoning vector as the mean of a Gaussian distribution. For each training sample $(x, y)$ with its corresponding latent reasoning vector $r$, we draw $K$ samples of reasoning. Formally, the $k$-th sampled reasoning ($k \neq 1$) is sampled as:

$$r^k = r + \epsilon, \quad \epsilon \sim \mathcal{N}(0, \sigma^2), \tag{5}$$

where $\mathcal{N}(0, \sigma^2)$ is a Gaussian distribution with zero mean and variance $\sigma^2$, $\epsilon$ denotes the sampled noise, and $\sigma$ can be thought of as a hyperparameter controlling the noise strength. Additionally, we include the original $r$ as the 1-st sample, *i.e.*, $r^1 = r$.

• **Reward Design.** After obtaining multiple latent reasoning samples in diverse directions, we evaluate which direction yields better outcomes by computing a reward for each. In the original GRPO, autoregressive decoding is used to generate the final output for each sample, and rewards are computed based on the quality of these outputs. However, this process is computationally expensive due to the need for full autoregressive decoding. To improve training efficiency, we instead propose using the model's perplexity (PPL) on the ground-truth answer as a proxy for the reward. This approach allows for faster evaluation.

Specifically, we construct the LLM input by combining the original prompt, each sampled latent reasoning vector, and the ground-truth answer. We then compute the model's perplexity for predicting the ground-truth tokens and use its negative value as the reward to evaluate the quality of each latent reasoning. Formally, the reward for the $k$-th sampled latent reasoning $r^k$ is defined as:

$$s^k = -exp(-\frac{1}{|y|}\sum_{i=1}^{|y|}log\pi_\theta(y_i|x, r^k, y_{<i})). \tag{6}$$

Here, $\pi_\theta$ represents the current policy model (*i.e.*, the current reasoning model), and $\pi_\theta(y_i|x, r^k, y_{<i})$ denotes the predicted probability for the $i$-th token in $y$. The rewards corresponding to the group of sampled reasoning instances are denoted as $S = [s^1, \ldots, s^K]$. Notably, the rewards are continuous rather than discrete.

• **Advantage Design.** After obtaining the reward scores, we compute the advantage. In the original GRPO, the advantage for each sampled reasoning instance is computed by comparing its reward to the average reward within the same sampling group. However, since our rewards are continuous, this approach may still yield positive advantages even when all samples in the group are of low quality, thus producing unreliable advantage signals. To mitigate this issue, we instead compute the advantage

by comparing each reward to the batch-level average reward. Specifically, for a sampled reasoning $r^k$ with reward score $s^k$, the advantage is computed as:

$$A^k = \frac{s^k - \bar{s}_{batch}}{\|\mathbf{S}_{batch} - \bar{s}_{batch}\|}, \qquad (7)$$

where $\mathbf{S}_{batch}$ denotes the batch of sampling groups, and $\bar{s}_{batch}$ represents the batch-level average reward, computed over the 1-st sample (i.e., the original reasoning before adding noise) from each group. Formally, it is given by $\bar{s}_{batch} = \frac{1}{N_{group}} \sum_{S=[s^1,\ldots,s^K] \in \mathbf{S}_{batch}} s^1$, where $N_{group}$ denotes the number of groups in this batch. Lastly, $\|\cdot\|$ denotes the L2 norm.

• **Policy Update.** After obtaining the advantage, we update our model in a way similar to GRPO. The optimization objective is formulated as follows:

$$L_{\text{GRPO}} = \sum_{(x,y) \in \mathcal{D}} -\frac{1}{K} \sum_{k=1}^{K} \frac{1}{|y|} \sum_{i=1}^{|y|} \frac{\pi_\theta(y_i \mid x, r^k, y_{<i})}{\pi_{\text{old}}(y_i \mid x, r^k, y_{<i})} \cdot A^k - \beta \mathbb{D}_{KL}(\pi_\theta \| \pi_{ref}). \qquad (8)$$

Notably, to reduce computational cost, we update only the "LatentRATT" layer while keeping the original LLM layers frozen. As a result, the second KL-divergence term, $\mathbb{D}_{KL}(\pi_\theta|\pi_{ref})$, becomes zero in the GRPO implementation.

## 4 EXPERIMENTS

In this section, we conduct experiments to answer the following research questions: **RQ1**: How does the overall recommendation performance of our proposed LatentR[3] method compare to existing recommendation methods? **RQ2**: Where do the improvements of LatentR[3] come from? **RQ3**: How do different design choices and reasoning hyperparameters affect the effectiveness of our method?

### 4.1 EXPERIMENTAL SETTINGS

**Dataset.** In line with prior LLM-based recommendation studies, we evaluate our method on several Amazon domain-specific datasets, including Toys, CDs, Games and Instruments. We adopt the standard preprocessing procedure that preserves chronological order for data filtering and splitting, with detailed steps and dataset statistics provided in Appendix A.

**Compared Methods.** We consider the top-$N$ recommendation task and compare our LatentR[3] with: (1) traditional sequential models, including **Caser** (Tang & Wang, 2018), **GRU4Rec** (Hidasi & Karatzoglou, 2018), and **SASRec** (Kang & McAuley, 2018); and (2) LLM-based approaches, including **Base** (direct LLM recommendation), **COT** (Tsai et al., 2024), **AlphaRec** (Sheng et al., 2025), **BIGRec** (Bao et al., 2025), and **D[3]** (Bao et al., 2024). AlphaRec is a state-of-the-art embedding-enhanced model, while D[3] is a state-of-the-art generative recommender. We implement LatentR[3] on both BIGRec and D[3]. Details on baselines and implementation are provided in Appendix B. Here, for explicit reasoning, we only include the direct CoT baseline. Other reasoning methods for recommendation mainly focus on rating prediction and require CoT supervision, so we exclude them. We have tried learning **explicit reasoning without CoT supervision via RL**, but it performed poorly (Appendix D), and we thus also omit it here. We also exclude general latent reasoning methods since they rely on CoT data, but include a no-RL variant in ablation study as their adaptation to our setting.

**Evaluation settings.** We evaluate the top-N recommendation effectiveness using Hit Ratio (HR@N) and Normalized Discounted Cumulative Gain (NDCG@N), with N set to 5 and 10. For brevity, we denote HR@5 and NDCG@5 as H@5 and N@5, respectively, similarly for '@10' cases.

### 4.2 MAIN RESULTS (RQ1)

Table 1 presents the overall performance comparison between our method and the baselines, including both traditional (Caser, GRU4Rec, SASRec) and LLM-based (w/o tuning: Base, COT; w/ tuning: AlphaRec, BIGRec, D[3]) approaches. From the results, we draw the following findings:

Table 1: Top-$N$ recommendation performance of LatentR$^3$ versus baselines. "RI." indicates the relative improvement of LatentR$^3$ on D$^3$ over each baseline (for the BIGRec column, "RI." uses LatentR$^3$ on BIGRec) across all datasets and metrics. Bold values denote the best results. The analysis of statistical significance is reported in the Appendix E, and our results are statistically significant.

| Dataset | Methods | Traditional | | | LLM-based | | | | | | |
|---|---|---|---|---|---|---|---|---|---|---|---|
| | Metrics | Caser | GRU4Rec | SASRec | Base | COT | AlphaRec | BIGRec | +LatentR$^3$ | D$^3$ | +LatentR$^3$ |
| Toys | H@5 | 0.0251 | 0.0417 | 0.0601 | 0.0203 | 0.0261 | 0.0579 | 0.0701 | 0.0821 | 0.0830 | **0.0898** |
| | H@10 | 0.0384 | 0.0564 | 0.0760 | 0.0359 | 0.0496 | 0.0893 | 0.0931 | 0.1107 | 0.1026 | **0.1152** |
| | N@5 | 0.0170 | 0.0305 | 0.0458 | 0.0128 | 0.0153 | 0.0347 | 0.0508 | 0.0600 | 0.0610 | **0.0670** |
| | N@10 | 0.0214 | 0.0352 | 0.0510 | 0.0178 | 0.0229 | 0.0448 | 0.0582 | 0.0693 | 0.0674 | **0.0752** |
| CDs | H@5 | 0.0469 | 0.0481 | 0.0841 | 0.0195 | 0.0302 | 0.0479 | 0.0757 | 0.0934 | 0.1122 | **0.1137** |
| | H@10 | 0.0689 | 0.0669 | 0.1054 | 0.0252 | 0.0406 | 0.0774 | 0.0929 | 0.1160 | 0.1272 | **0.1327** |
| | N@5 | 0.0312 | 0.0365 | 0.0622 | 0.0148 | 0.0213 | 0.0278 | 0.0616 | 0.0754 | 0.0906 | **0.0915** |
| | N@10 | 0.0382 | 0.0425 | 0.0691 | 0.0167 | 0.0246 | 0.0373 | 0.0672 | 0.0826 | 0.0955 | **0.0977** |
| Games | H@5 | 0.0324 | 0.0322 | 0.0416 | 0.0236 | 0.0120 | 0.0558 | 0.0461 | 0.0580 | 0.0608 | **0.0716** |
| | H@10 | 0.0538 | 0.0517 | 0.0633 | 0.0311 | 0.0194 | 0.0893 | 0.0709 | 0.0870 | 0.0860 | **0.1006** |
| | N@5 | 0.0211 | 0.0207 | 0.0280 | 0.0190 | 0.0082 | 0.0397 | 0.0334 | 0.0413 | 0.0423 | **0.0507** |
| | N@10 | 0.0280 | 0.0270 | 0.0350 | 0.0214 | 0.0105 | 0.0515 | 0.0414 | 0.0506 | 0.0505 | **0.0601** |
| Instruments | H@5 | 0.0781 | 0.0766 | 0.0793 | 0.0154 | 0.0135 | 0.0813 | 0.0938 | 0.1029 | 0.0984 | **0.1066** |
| | H@10 | 0.0977 | 0.0960 | 0.0950 | 0.0192 | 0.0199 | 0.1051 | 0.1158 | 0.1214 | 0.1167 | **0.1229** |
| | N@5 | 0.0564 | 0.0630 | 0.0708 | 0.0296 | 0.0261 | 0.0564 | 0.0807 | 0.0882 | 0.0848 | **0.0920** |
| | N@10 | 0.0627 | 0.0692 | 0.0758 | 0.0411 | 0.0452 | 0.0640 | 0.0879 | 0.0941 | 0.0907 | **0.0973** |
| | RI | 85.8% | 67.8% | 27.9% | 266.8% | 245.9% | 38.8% | 17.0% | - | 8.4% | - |

- The best version of our method (LatentR$^3$ applied to D$^3$) demonstrates superior performance compared to all existing approaches across all metrics on all three datasets, consistently validating the effectiveness of our proposed methodology.

- Focusing on the comparison among the LLM-based methods with recommendation-specific tuning, those that use LLMs as recommenders (BIGRec and D$^3$) outperform the method that leverages LLMs to enhance traditional models (AlphaRec) on the Toys, CDs, and Instruments, but underperform on Games. After applying our LatentR$^3$, the performance of both BIGRec and D$^3$ improves significantly, with BIGRec achieving a relative improvement (RI) of 17.0% and D$^3$ achieving an RI of 8.4%. Furthermore, the enhanced models surpass AlphaRec. These results show that 1) incorporating latent reasoning substantially improves the performance of the LLM-based methods; 2) our latent reasoning method can be applied to different existing LLM-based methods.

- Directly using LLMs for recommendation without tuning (Base) performs substantially worse than traditional methods. Adding explicit reasoning via CoT provides slight improvements but still largely falls short of traditional baselines. After tuning, the best-performing LLM-based baselines outperform traditional models in nearly all cases. These results underscore the importance of explicitly aligning LLMs—including their reasoning capabilities—with the recommendation task.

To further validate our method, Appendix F compares LatentR$^3$ with BIGRec on a larger LLM, showing consistent gains, and Appendix C provides additional results on a non-Amazon dataset, where our method continues to achieve better performance.

**Efficiency Comparison:** Beyond accuracy, a big advantage of our method is that it needs only a few latent tokens (one in our experiments) to represent reasoning, greatly reducing inference latency. An empirical study on four datasets (Appendix G) shows that our method maintains a cost close to non-reasoning LLM baselines, whereas explicit COT incurs a substantial increase (*e.g.*, a 25-fold increase on Toys and nearly a 30-fold increase on Games).

## 4.3    In-depth Analysis (RQ2 & RQ3)

In this section, we first analyze the performance improvements of our method across different item groups to understand where the gains originate. We then investigate the impact of various factors on the method's effectiveness, starting with ablation studies to assess the contribution of each design component, followed by an analysis of the reasoning length. Lastly, we compare our modified GRPO with the original GRPO method. To reduce the computational cost (RL), we limit our analysis to the Toys and CDs datasets, except for the study analyzing the source of performance gains. Besides, we also study the Applicability of our method to other models in Appendix M.

Table 2: Ablation study results on the Toys and CDs datasets. The best results are highlighted in bold.

| Method | Toys | | | | CDs | | | |
|---|---|---|---|---|---|---|---|---|
| | H@5 | H@10 | N@5 | N@10 | H@5 | H@10 | N@5 | N@10 |
| LatentR$^3$ | **0.0821** | **0.1107** | **0.0600** | **0.0693** | **0.0934** | **0.1160** | **0.0754** | **0.0826** |
| w/o Reasoning | 0.0701 | 0.0931 | 0.0508 | 0.0582 | 0.0757 | 0.0929 | 0.0616 | 0.0672 |
| w/o LatentRATT | 0.0772 | 0.1040 | 0.0574 | 0.0661 | 0.0705 | 0.0875 | 0.0568 | 0.0624 |
| w/o RL (only SFT) | 0.0804 | 0.1067 | 0.0584 | 0.0669 | 0.0830 | 0.1005 | 0.0662 | 0.0719 |
| w/o Batch Advantage | 0.0812 | 0.1083 | 0.0589 | 0.0676 | 0.0828 | 0.1002 | 0.0661 | 0.0718 |

### 4.3.1 PERFORMANCE OVER POPULAR AND UNPOPULAR ITEMS

For latent reasoning, it is difficult to directly demonstrate why it leads to performance improvements, as can be done with explicit CoT. Therefore, we consider providing indirect evidence. Intuitively, reasoning should be more beneficial for the more challenging aspects of recommendation, where it is expected to deliver greater gains. Long-tail (i.e., unpopular) items are typically more difficult to recommend accurately, while existing

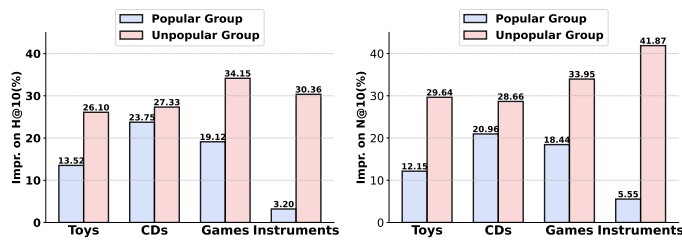

Figure 2: Performance improvement of LatentR$^3$ over BIGRec on both popular and unpopular items.

methods already perform well on popular items. To test this hypothesis, we examine whether our method yields greater improvements on unpopular items. Specifically, we compare the relative performance gains of our method over BIGRec across items with different popularity levels. The results are shown in Figure 2 (see Appendix H for more results and experimental settings). As shown, our method achieves significantly larger improvements on unpopular items, suggesting that the incorporation of reasoning is particularly beneficial in more challenging recommendation scenarios.

### 4.3.2 ABLATION STUDY

Our method features key innovations in both the model architecture and the learning algorithm. To assess the contribution of each component, we conduct ablation studies on both aspects. Specifically, we evaluate the following variants of our method: 1) w/o Reasoning, which removes the latent reasoning component entirely. 2) w/o LatentRATT, which omits the LatentRATT layer and directly uses the final hidden states from the LLM to construct the latent reasoning token; 3) w/o RL, which trains the reasoning model using only SFT without RL; 4) w/o Batch Advantage, which replaces our batch-based advantage estimation with the original GRPO setting that computes advantages with a group average reward. Specifically, variants 1) w/o Reasoning, 2) w/o LatentRATT, and 3) w/o RL are fully fine-tuned, while variant 4) w/o Batch Advantage is fully fine-tuned during SFT but only the LatentRATT layer is tuned during RL. Notably, regarding the reward design, we would study it later when compared to the original GRPO. We conduct experiments on the Toys and CDs datasets.

The comparison results are summarized in Table 2. From the architectural perspective, removing the LatentRATT layer leads to a significant performance drop. On the CDs dataset, this variant even underperforms the "w/o Reasoning" version, underscoring the importance of LatentRATT in generating meaningful and space-aligned reasoning outputs. From the learning algorithm side, the "w/o RL" variant—which only uses SFT—performs better than the "w/o Reasoning" version, indicating that SFT can partially enable latent reasoning. However, its performance remains inferior to the full method with RL, confirming that effective latent reasoning learning relies on reinforcement training. Furthermore, replacing the batch-based advantage with the original GRPO's advantage (as in "w/o Batch Advantage") results in comparable or worse performance than the "w/o RL" version, demonstrating the necessity of batch-based reward estimation for stable RL learning under our reward

design. Overall, these findings highlight the critical roles of both our architectural and learning algorithm designs in the method's effectiveness. See Appendix I for training-time analysis.

### 4.3.3 INFLUENCE OF REASONING LENGTH

We next investigate how the length of latent reasoning affects model performance by varying the number of generated latent tokens ($K$). Considering the cost of RL tuning, we just tune $K$ in {0,1,2}, and conducted experiments on Toys and CDs. The results in terms of NDCG@5 (N@5) and HR@5 (H@5) are shown in Figure 3. Results for other metrics and the detailed implementations can be found in Appendix K. As shown, increasing the reasoning length improves performance, demonstrating the potential to achieve even better results by increasing the reasoning length. However, the gain from $K$=1 to $K$=2 is much smaller than from $K$=0 to $K$=1, suggesting that a few latent tokens may be sufficient to capture most of the reasoning information.

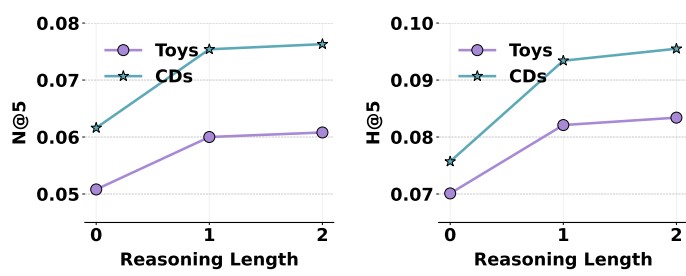

Figure 3: Impact of Reasoning Length on LatentR$^3$ Performance.

### 4.3.4 COMPARISON WITH THE ORIGINAL GRPO

We also compare our LR-GRPO with the original GRPO, which relies on generating complete answers for reward computation for group sampling, to evaluate its effectiveness. To improve training efficiency, we propose using the model's perplexity (PPL) on the ground-truth answer as a proxy for the reward. This yields a continuous reward signal and motivates adjustments to the advantage computation. To assess the impact of these modifications, we implement a variant of our method using the original GRPO. The comparison results on the

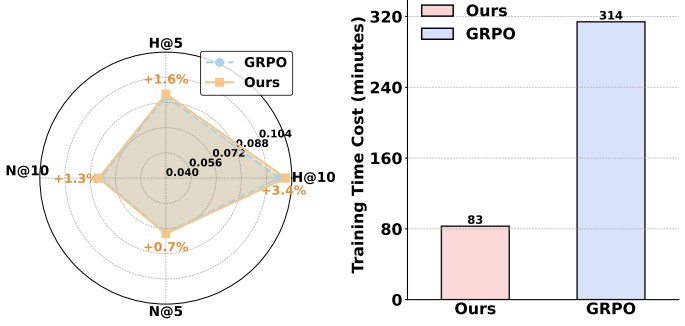

Figure 4: Comparison to the original GRPO regarding Performance (Left) and efficiency (Right).

CDs dataset are shown in Figure 4, with additional results and implementation details provided in Appendix L. In terms of performance, our method achieves results comparable to the variant using original GRPO—slightly outperforming it on CDs while performing slightly worse on Toys (see Appendix L). In terms of training efficiency, our method is significantly faster, reducing the training cost to approximately 1/4 of that of the original GRPO. These findings validate the rationale of our modifications.

## 5 RELATED WORK

Our work relates to LLM-based recommendation and latent reasoning. Due to space limits, we briefly cover LLM-based recommendation and reasoning methods here; a full review is in Appendix N. Existing works include three main LLM-based recommendation paradigms: (1) in-context learning (Gao et al., 2023; Sun et al., 2024; Wang et al., 2021), (2) agent-based frameworks (Wang et al., 2023; Zhang et al., 2024a;b; Shi et al., 2024), and (3) fine-tuning-based methods (Bao et al., 2025; Zhang et al., 2024c). Among these, the fine-tuning-based solution has received the most

attention due to its ability to more effectively align LLMs with the task, and our work falls into this category. In this direction, only a few studies have explored reasoning, with existing works primarily aiming to enhance explicit CoT reasoning via fine-tuning, such as RecSAVER (Tsai et al., 2024), ReasoningRec (Bismay et al., 2025), Reasoning4Rec (Fang et al., 2025), $R^2$ec (You et al., 2025), and Exp3rt (Kim et al.). However, these approaches rely heavily on explicit CoT, which is costly to obtain and typically leads to high inference latency. In contrast, our work introduces latent reasoning for LLM-based recommendation, which eliminates the need for CoT supervision and enables more efficient inference. As for latent reasoning, only two concurrent works (Tang et al., 2025; Zhang et al., 2025) have explored it, but neither focuses on LLM-based recommendation, and our approach to latent reasoning is fundamentally different and specifically focused on LLM-based recommendation.

## 6 CONCLUSION

In this work, we proposed LatentR$^3$, a novel LLM-based recommendation framework that replaces explicit chain-of-thought reasoning with compact latent reasoning representations. By integrating architectural innovations such as the LatentRATT layer with a two-stage reinforcement learning strategy, LatentR$^3$ enables efficient and effective preference reasoning without relying on costly CoT supervision. Extensive experiments show that our method significantly enhances the performance of existing LLM-based recommendation models, while introducing only minimal additional inference cost—equivalent to generating a single extra token. These results highlight the potential of latent reasoning as a practical and scalable alternative for LLM-based recommendation.

**Limitations and Future Works.** Following common practices for LLM-based recommendation works, our current experiments are limited to relatively small datasets due to the very high cost. In the future, we plan to explore the effectiveness of our method on diverse, large-scale datasets. Additionally, compared to explicit CoT, latent reasoning offers less interpretability. We plan to investigate ways to improve the explainability of latent reasoning, facilitating its wider adoption.

## ETHICS STATEMENT

This work adheres to the ICLR Code of Ethics. Our research does not involve human subjects or sensitive personal data. Although we use some user interaction data, it is anonymized, publicly available, and does not contain any personally identifiable information. Moreover, we strictly follow the usage guidelines set by the dataset providers. However, when applying our method to broader scenarios in the future, it would be advisable to include additional user privacy protections, such as allowing users to control how their data is used and extending the approach to local learning settings. The experiments presented in this paper focus on algorithmic development and theoretical analysis, and do not entail any direct applications that could cause harm if misused. All authors have disclosed any potential conflicts of interest, and there are no financial or institutional relationships that may influence the objectivity of this research. The usage of LLMs is provided in Appendix O.

## REPRODUCIBILITY STATEMENT

To support reproducibility, we provide a detailed description of all experimental settings, model architectures, and hyperparameters in Appendix B. Our code has been made available at https://github.com/xuwenxinedu/R3, including training scripts, evaluation scripts, and instructions for reproducing the main results. All datasets used are publicly accessible, and we provide the dataset preprocessing process in Appendix A and the preprocessing script in our code.

### ACKNOWLEDGMENTS

This research is supported by the National Natural Science Foundation of China (U24B20180). This research is supported by A*STAR under its Japan-Singapore Joint Call: Japan Science and Technology Agency (JST) and Agency for Science, Technology and Research (A*STAR) 2024 (Award R24I6IR142). Any opinions, findings and conclusions or recommendations expressed in this material are those of the author(s) and do not reflect the views of the A*STAR.

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

## A  DATASET PRE-PROCESSING

Our experiments were conducted on data from Amazon, with primary evaluations performed on three domain-specific datasets: Toys, CDs, Games and Instruments. Following established methodology, we preprocessed each subset to achieve 5-core data integrity. To address the computational demands of LLM training while maintaining dataset validity, we implemented a dynamic temporal partitioning strategy: We iteratively process the dataset within a sliding time window, starting from October 2017 to October 2018. After applying 5-core filtering, if the resulting item count falls below 5,000, the start time is shifted back by 3 months (e.g., to July 2017). This adjustment repeats until the processed dataset contains more than 5,000 items, with the end time fixed at October 2018 throughout the process. Key dataset statistics are summarized in the accompanying table3. To align with real-world scenarios, we adopt a temporal split of the processed datasets into training, validation, and testing sets based on timestamps, following an 8:1:1 ratio. In alignment with comparative baseline requirements across all experimental conditions, we standardized the maximum sequence length at 10 for consistency in temporal pattern analysis.

Table 3: Dataset statistics.

| Dataset | Train | Valid | Test | Item |
|---|---|---|---|---|
| Toys | 53898 | 6737 | 6738 | 6299 |
| CDs | 49251 | 6156 | 6158 | 5841 |
| Games | 75175 | 9397 | 9397 | 5308 |
| Instruments | 66500 | 8312 | 8313 | 5030 |

## B COMPARED METHODS AND IMPLEMENTATION DETAILS

• **Baselines.** We consider the top-N recommendation task. To evaluate the effectiveness of our method, we compare it against both representative traditional sequential recommendation models and state-of-the-art (SOTA) LLM-based recommendation approaches:

- **Caser** (Tang & Wang, 2018): This is a well-known sequential recommendation approach that utilizes Convolutional Neural Networks (CNNs) to capture sequential patterns and model user preferences.

- **GRU4Rec** (Hidasi & Karatzoglou, 2018): This is another widely recognized method that leverages Gated Recurrent Units (GRUs) to encode sequential patterns and model user preferences.

- **SASRec** (Kang & McAuley, 2018): This is a highly representative sequential recommendation method that utilizes a self-attention network to model user preferences, with an architecture resembling that of the decoder-only Transformer model.

- **Base**: This refers to the method that directly prompts the backbone LLM to perform the recommendation task.

- **COT** (Tsai et al., 2024): This refers to the zero-shot version of Rec-SAVER, which uses CoT reasoning to prompt the LLM for recommendation, *i.e.*, instructing the LLM to produce a reasoning process before generating the final results.

- **AlphaRec** (Sheng et al., 2025): This is an SOTA LLM-based method that leverages LLM-generated embeddings to enhance the recommendation model.

- **BIGRec** (Bao et al., 2025): This is a representative LLM-based generative recommendation method that fine-tunes LLMs to generate next-item predictions, with specific designs to support full ranking.

- **$D^3$** (Bao et al., 2024): This is a SOTA LLM-based generative recommendation method. It follows a similar fine-tuning process to BIGRec but differs in its inference strategy. Specifically, it introduces debiasing techniques during inference to enhance the quality of generated recommendations. Additionally, it includes an ensemble design with traditional models; however, we omit this component in our implementation to ensure a fair comparison.

Our method, LatentR$^3$, is compatible with various LLM-based generative recommendation approaches. Accordingly, we implement it on both BIGRec and $D^3$. Notably, existing LLM reasoning methods for recommendation cannot be selected for comparison because: (1) they are not designed for top-N recommendation tasks, and (2) their training relies on explicit CoT supervision, which is difficult to obtain in our setting. To align with our setting, we also attempted to learn explicit CoT reasoning directly via RL; however, the results were very poor, as shown in Appendix D. Similarly, we exclude general-domain latent reasoning methods for comparison, as they still depend on CoT data for learning. However, in our ablation study, we will include a variant of our method without RL training, which can be viewed as an adaptation of the general latent reasoning method to our setting.

• **Implementation Details** For traditional recommendation models, we use the Adam optimizer and perform grid searches over learning rates 1e-2, 1e-3, 1e-4 and weight decay values 1e-4, 1e-5, 1e-6, and all models are trained using Binary Cross-Entropy loss with randomly sampled negative items. For LLM-based methods, we use Qwen2.5-1.5B Team (2024) as the backbone LLM. Supervised fine-tuning (SFT) is conducted using the AdamW optimizer, with learning rates selected from {3e-3, 3e-4, 3e-5}, and early stopping is applied with a patience of 1. During the reinforcement learning stage, we search learning rates within {1e-5, 1e-4, 5e-4}. All experiments are run on 2 NVIDIA A100 GPUs.

Notably, BIGRec originally uses a grounding-based (matching) method for decoding items. In our implementation, we adopt a constrained decoding approach, as we found it yields better performance. This is supported by the empirical results on the CDs dataset in Table 4.

Table 4: Performance of BIGRec using different decoding methods—(1) Grounding and (2) Constrained—on the CDs dataset.

| Decoding method | H@5 | H@10 | N@5 | N@10 |
|---|---|---|---|---|
| Grounding Decoding | 0.0609 | 0.0670 | 0.0573 | 0.0592 |
| Constrained Decoding | 0.0757 | 0.0929 | 0.0616 | 0.0672 |

## C    RESULTS ON DATASET BEYOND AMAZON

We have included a new dataset, Steam, which contains about 108K users, 15K items, and 314K interactions. The results are summarized in the Table 5, and our method continues to deliver substantial performance improvements. Notably, many recent ID-based methods (Hou et al., 2025) are also evaluated on datasets with item counts similar to those in this experiment. As shown in the table, our method could still achieve large performance improvements compared to baselines.

## D    LEARNING EXPLICIT REASONING WITHOUT COT SUPERVISION VIA RL

We also experimented with directly optimizing CoT generation via RL, similar to DeepSeek-R1-zero, denoted as "COT-RL." We compare it with the following methods: Base, COT, and our LatentR$^3$. The results are summarized in Table 6. As shown, both COT and COT-RL outperform the base LLM. However, applying RL to learn CoT reasoning without direct supervision fails to improve performance and can even degrade it. This is likely due to the inherent difficulty of learning long CoT chains: correct sampling is challenging, and the learning process requires considerations beyond accuracy—such as output formatting—which can reduce learning efficiency and lead to reward increases without corresponding performance gains.

## E    THE ANALYSIS OF STATISTICAL SIGNIFICANCE

In the main result (Table 1), we disabled the sources of randomness to ensure reproducibility. To address your concern, we enabled random sampling during generation and conducted a t-test between our method and the corresponding baseline (BIGRec). When reporting the averaged results, our method remains significantly better, with a p-value < 0.001, as shown in Table 7.

## F    PERFORMANCE ON LARGER LLM

We have further explored the effectiveness of our method on a larger LLM, Qwen2.5-3B. Considering the high cost of LLM tuning, we focused on comparing our method with BIGRec on only one dataset — CDs. Meanwhile, given the much higher cost of RL tuning for larger LLMs, we apply LoRA-based tuning to control training expenses under the use of larger LLMs. The results are summarized in Table 8. As shown, our method still delivers a more substantial relative improvement on the larger model, achieving an average performance gain of 40.7%.

## G    INFERENCE COST COMPARISON

Compared to non-reasoning LLM baselines, our method introduces the extra cost of generating latent reasoning tokens. However, since only a few tokens are used (one by default), the added cost is minimal. In contrast, explicit reasoning methods often require a large and uncontrollable number of tokens, leading to significant overhead. We randomly selected 100 samples from each of the four datasets. Inference was performed using a single A100 GPU, with batch size set to 4 and beam size

Table 5: Performance of LatentR$^3$ and baselines on the Steam dataset.

| Method | H@5 | H@10 | N@5 | N@10 |
|---|---|---|---|---|
| Caser | 0.0407 | 0.0685 | 0.0262 | 0.0351 |
| GRU4Rec | 0.0422 | 0.0721 | 0.0271 | 0.0367 |
| SASRec | 0.0390 | 0.0633 | 0.0251 | 0.0329 |
| BIGRec | 0.0559 | 0.0821 | 0.0405 | 0.0489 |
| LatentR$^3$ | **0.0630** | **0.0959** | **0.0451** | **0.0557** |

Table 6: Performance comparison of explicit reasoning learned via RL (without CoT supervision, denoted by "COT-RL", ) versus other methods.

| Metrics | H@5 | H@10 | N@5 | N@10 |
|---|---|---|---|---|
| Base | 0.0195 | 0.0252 | 0.0148 | 0.0167 |
| COT | 0.0302 | 0.0406 | 0.0213 | 0.0246 |
| COT-RL | 0.0297 | 0.0390 | 0.0202 | 0.0232 |
| Ours | 0.0934 | 0.1160 | 0.0754 | 0.0826 |

set to 10 (the CoT method set to 1). Each measurement was taken three times and we report the average values, as shown in the Figure 5. Because only a single token is added, and the length of item titles inherently varies, our inference time is almost identical to that of non-reasoning methods. However, explicit CoT methods are far more expensive (COT).

## H PERFORMANCE ON POPULAR AND UNPOPULAR ITEMS

Based on the frequency of items in the training set, we consider the top 20% of items as popular items and the other items as unpopular items. We then calculate the performance of BIGRec and LatentR$^3$ for these two groups of items, and calculate the relative performance improvement in each group. The specific results are shown in Figure 6.

To further validate our method's ability to recommend unpopular items, we added SASRec results and compared the performance improvements of BIGRec and LatentR$^3$ over SASRec. The results on the Toys dataset are shown in Table 9, where the absolute results N@10 are also included. As shown in the table below, for the Toys dataset, BIGRec achieves comparable relative improvements over SASRec for long-tail and popular items (17.4% vs. 13.7%). In contrast, LatentR3 shows a significantly larger relative improvement for long-tail items compared to popular items (53% vs.

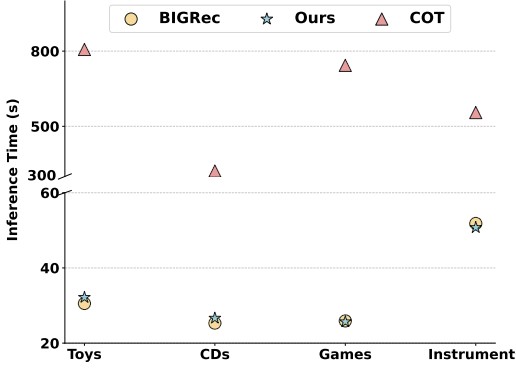

Figure 5: Inference time comparison across non-reasoning (BIGRec), LatentR$^3$ (Ours), and explicit CoT methods (COT).

Table 7: T-test results for BIGRec and LatentR$^3$ (applied to BIGRec) on the CDs dataset over five experimental runs. "*" indicates statistical significance with $p < 0.01$.

| Method | H@5 | H@10 | N@5 | N@10 |
|---|---|---|---|---|
| BIGRec | 0.0725 | 0.0902 | 0.0595 | 0.0653 |
| +LatentR$^3$ | 0.0929* | 0.1131* | 0.0748* | 0.0813* |
| p-value | 0.00014 | 0.00067 | 4.8e-6 | 1.7e-5 |

Table 8: Performance comparison of BIGRec and LatentR$^3$ on the CDs dataset using larger LLMs with LoRA tuning.

| Method | H@5 | H@10 | N@5 | N@10 | Impr. |
|---|---|---|---|---|---|
| BIGRec | 0.0503 | 0.0658 | 0.0389 | 0.0439 | - |
| +LatentR$^3$ | 0.0695 | 0.0903 | 0.0566 | 0.0634 | 40.7% |

27.5%). This indicates that while BIGRec's gains over SASRec are similar across item popularity levels, LatentR$^3$ benefits much more on long-tail items.

Besides, we also explored the performance of our method on the fully cold-start items, with the results shown in Table 10. As the results show, our method also has better cold-start performance than the BIGRec baseline.

## I    MORE ABLATION RESULTS

Our RL designs include mainly three parts: 1) batch-level advantage, 2)the sampling change, and 3) the PPL-based rewards. The batch-level design aims to improve performance, while the PPL-based rewards aim to enhance efficiency. The change in sampling arises from the fact that we can no longer follow the sampling procedure used in the GRPO paper. Reparameterization-based sampling is a straightforward choice for handling latent reasoning. Even when implementing the original GRPO under latent reasoning, we still need to adopt our sampling method. Here, we additionally verify the functions of different designs, reporting the time cost. The results are shown in Table 11. As shown, the performance improvement is mainly attributed to the batch-level advantage, and this batch-level advantage does not contribute to efficiency improvements. Please note that, in terms of time efficiency, our method is more efficient — it takes only about one quarter of the runtime of the original GRPO (under the changed sampling), as reported in Figure 4 of our paper.

## J    ABLATION STUDY ON REASONING TOKENS

To isolate and demonstrate that the observed gain is primarily due to the learned reasoning captured by the latent tokens rather than merely an enhanced utilization of the LLM's semantic information, we conduct a specific ablation study. We mask the reasoning tokens generated by the latent reasoning module during inference (denoted as "+ mask token"), effectively removing the model's capacity to perform reasoning while retaining access to the LLM's semantic item representations. The results in Table 12 show that masking the reasoning tokens leads to a substantial performance drop, confirming the importance of the reasoning component.

## K    INFLUENCE OF REASONING LENGTH

We investigate how length of latent reasoning affects model perfomance by varying the number of generated latent tokens ($K$), with $K \in \{0, 1, 2\}$. When implementing, a model dedicated to single reasoning token generation was fully trained. We directly perform reinforcement learning to train the 2 reasoning tokens model based on this. Noise sampling was selectively applied only to the last reasoning token, given that the first token's generative capabilities were already robustly

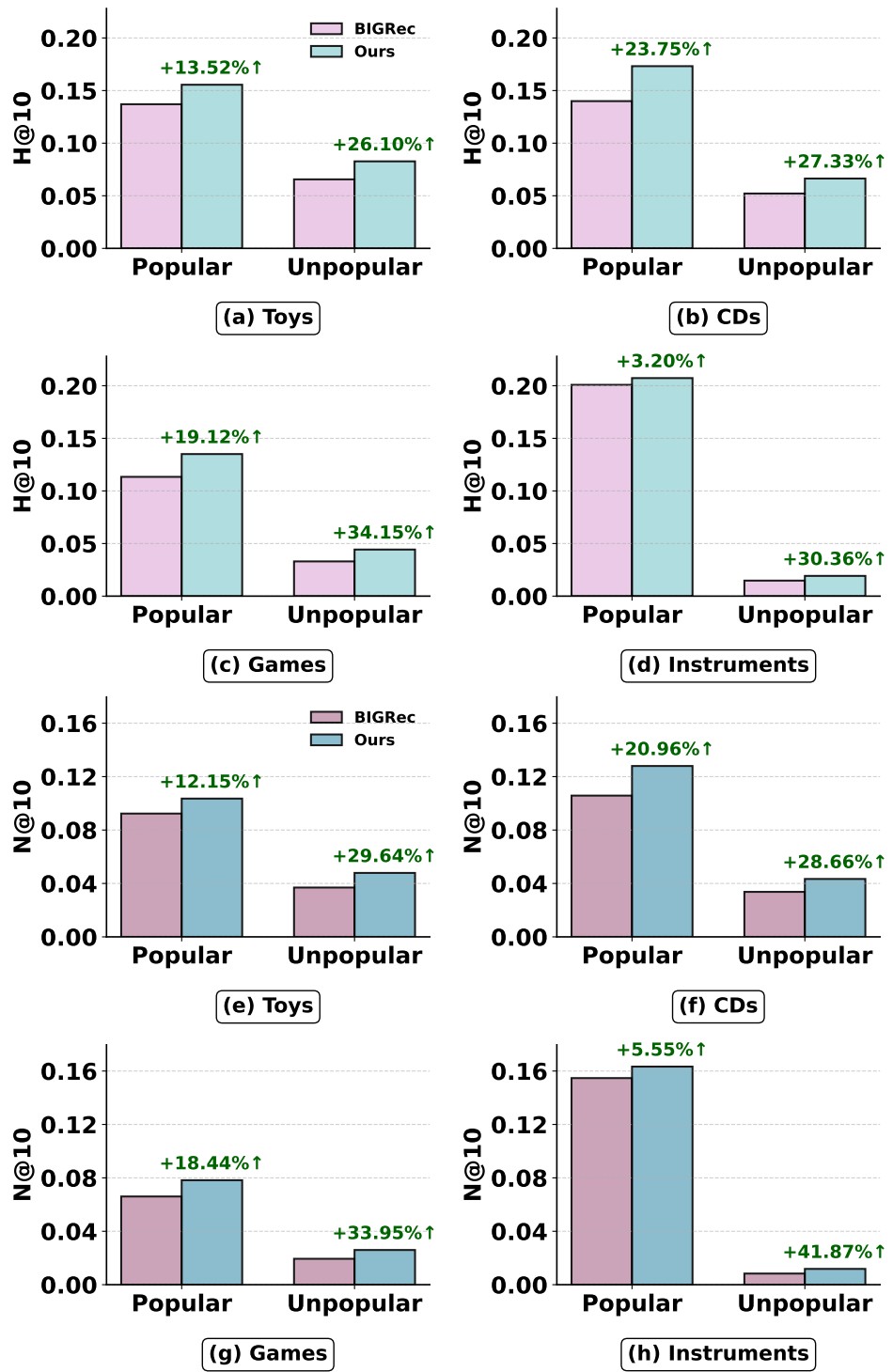

Figure 6: Performance improvement of LatentR[3] over BIGRec on both popular and unpopular items on four dataset.

Table 9: Relative performance improvement comparison between popular and unpopular items

| Toys | Pupular | RI | Long-tail | RI |
|---|---|---|---|---|
| SASRec | 0.0812 | - | 0.0315 | |
| BIGRec | 0.0923 | **13.7%** | 0.0369 | **17.14%** |
| LatentR$^3$ | 0.1035 | **27.5%** | 0.0479 | **52.1%** |

Table 10: Performance on the fully cold-start items.

| Dataset | Toys | CDs |
|---|---|---|
| BIGRec | 0 | 0 |
| LatentR$^3$ | 0.0029 | 0.0263 |

established through the initial training. Experiments are conducted on the Toys and CDs datasets, and the results in terms of NDCG@10 (denoted as N@10), HR@10 (H@10), NDCG@5 (N@5) and HR@5 (H@5) are shown in Figure 7. As shown, increasing the reasoning length improves performance, demonstrating the potential to achieve even better results by increasing the reasoning length. However, the gain from $K$=1 to $K$=2 is much smaller than from $K$=0 to $K$=1, suggesting that a few latent tokens may be sufficient to capture most of the reasoning information.

## L    COMPARISON WITH THE ORIGINAL GRPO

To implement the variant with the original GRPO, after sampling the latent reasoning, we leverage the LLM to generate the final answer, and then exactly match the answer with the ground-truth to compute 0-1 reward. Lastly we leverage the group average for advantage computation. The experiments are conducted on both the Toys and CDs datasets. The results are summarized in Figure 8. In terms of performance, our method achieves results comparable to the variant using original GRPO—slightly outperforming it on the CDs dataset while performing slightly worse on the Toys dataset. In terms of training efficiency, our method is significantly faster, reducing the training cost to approximately 1/4 of that of the original GRPO-based method. These findings validate the rationale and effectiveness of our proposed modifications. Notably, the reported results for the original GRPO are based on full-model tuning during RL, as this setting is necessary to achieve the reported performance. Tuning only the LatentRATT layer, as done in the default LatentR$^3$—results in significantly worse outcomes for GRPO. However, we observe that limiting GRPO to tuning the LatentRATT layer would only slightly reduce training cost (by approximately 50 minutes), indicating that our method remains substantially more efficient.

## M    APPLICABILITY OF OUR METHOD TO OTHER MODELS

In this section, we have explored two broader applications of methods: 1) applying to traditional ID-based recommendation, and 2) applying to the method that leverages LLM-generated embeddings to enhance the recommendation model (AlphaRec).

**Applying to SASRec**: We applied our method to SASRec, and the results are reported in Table 13. As shown, SASRec does not exhibit meaningful gains after incorporating our method. These results align with the findings from ReaRec's Tang et al. (2025) ablation studies — applying LLMs' latent reasoning into traditional methods without specifically solving problems like reasoning degradation does not lead to effective improvements. However, our LatentR$^3$ could bring improvements over the vanilla latent reasoning methods.

**Applying to Enhance LLM Representations**: To verify whether latent reasoning can lead to better LLM representations, we conducted additional experiments on AlphaRec, replacing the standard embeddings with those generated by the our latent reasoning model to enhance the ID-based model. The results presented in Table 14 demonstrate that the latent reasoning approach also exhibits considerable value in this scenario.

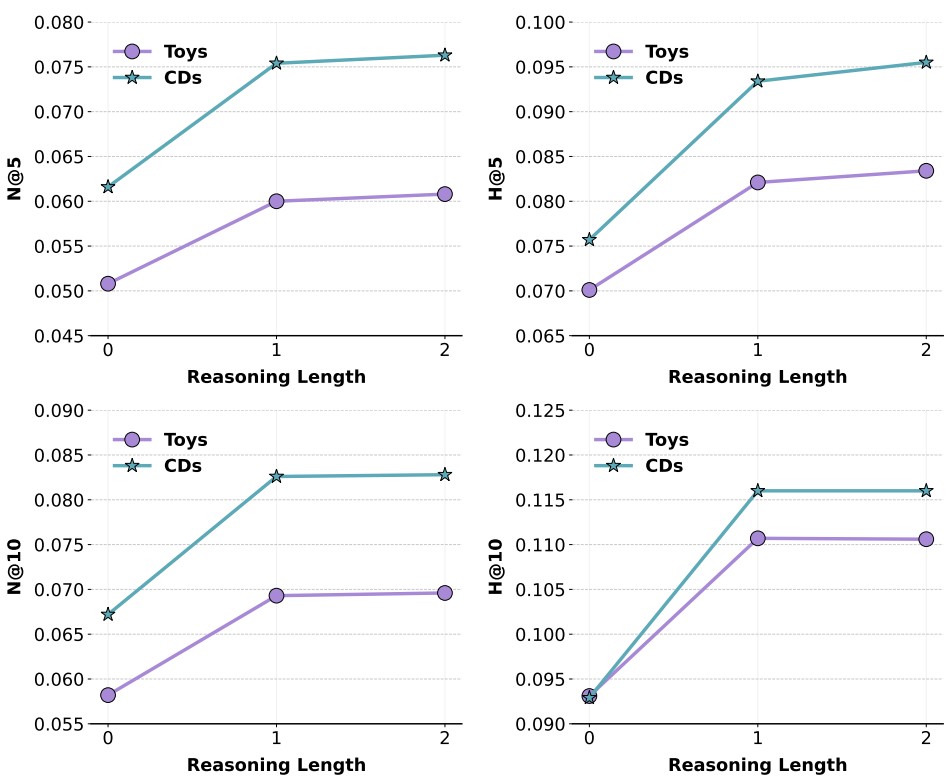

Figure 7: Impact of Reasoning Length on LatentR$^3$ Performance

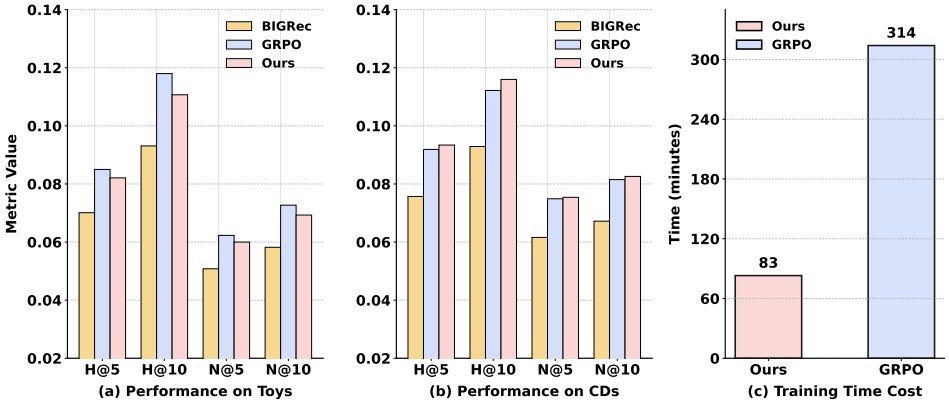

Figure 8: Comparison to the original GRPO regarding Performance (Left) and efficiency (Right).

Table 11: Detailed comparison of different designs

| Ablations | H@10 | Train time |
|---|---|---|
| +sampling+PPL rewards (i.e., w/o Batch Advantage) | 0.1002 | 76 mins |
| +sampling + PPL rewards+ batch-level advantage (i.e., LatentR3) | 0.1160 | 81 mins |

Table 12: Ablation study on reasoning tokens

| | H@5 | H@10 | N@5 | N@10 |
|---|---|---|---|---|
| BIGRec | 0.0616 | 0.0672 | 0.0757 | 0.0929 |
| LatentR$^3$ | 0.0754 | 0.0826 | 0.0934 | 0.1160 |
| + mask token | 0.0644 | 0.0707 | 0.0833 | 0.1031 |

## N  FULL RELATED WORK

**LLM-based Recommendation.**    The recent widespread success of LLMs (Zhao et al., 2026; Team, 2024; Zhang et al., 2023) has sparked growing interest in their application to recommendation systems, giving rise to three main paradigms: (1) in-context learning (Gao et al., 2023; Sun et al., 2024; Wang et al., 2021), (2) agent-based frameworks (Wang et al., 2023; Zhang et al., 2024a;b; Zhao et al., 2025), and (3) fine-tuning-based methods (Bao et al., 2025; Zhang et al., 2024c; Lin et al., 2025b; Gao et al., 2025). Among these, the fine-tuning-based solution has received the most attention due to its ability to more effectively align LLMs with recommendation objectives, and our work falls into this category. Although significant efforts have been devoted to this direction, most existing methods overlook the role of reasoning in recommendation—particularly latent reasoning, which remains unexplored. Among the limited studies that investigate reasoning, existing works primarily focus on enhancing explicit chain-of-thought (CoT) reasoning through fine-tuning. RecSAVER (Tsai et al., 2024) and ReasoningRec (Bismay et al., 2025) leverage larger LLMs to generate CoT reasoning data for training smaller LLMs, aiming to improve their reasoning capabilities. Reasoning4Rec (Fang et al., 2025) decomposes the reasoning process into multiple steps, using user reviews as proxy supervision. Exp3rt (Kim et al.) focuses on distilling the reasoning behavior of larger LLMs into smaller models. However, these approaches rely heavily on CoT data, which is costly to obtain and often difficult to ensure in quality. Moreover, the use of explicit CoT reasoning typically incurs higher inference latency. In contrast, our work introduces latent reasoning for LLM-based recommendation, which eliminates the need for CoT supervision and enables more efficient inference. Rec-R1 Lin et al. (2025a) investigated the use of RL to enhance LLM-based recommendation; however, it focuses on enhancing query rewriting and summarization of LLMs rather than improving general reasoning capabilities. As for latent reasoning, only two concurrent works (Tang et al., 2025; Zhang et al., 2025) have explored it, but neither focuses on LLM-based recommendation, and our approach to latent reasoning is fundamentally different and specifically focused on LLM-based recommendation. ReaRec Tang et al. (2025) does not focus on reinforcement learning to support better exploration of reasoning. STREAM-Rec focuses more on building a recommendation foundation model with reasoning abilities from scratch. While STREAM-Rec Zhang et al. (2025) uses RL, it still relies on pseudo labels to train slow-thinking abilities, and its RL stage largely follows the original GRPO design, except that its reward computation incorporates recommendation-specific elements. Our sampling and advantage computation are different.

Table 13: The results of replacing LLMs with SASRec for LatentR$^3$.

| Method | H@5 | H@10 | N@5 | N@10 |
|---|---|---|---|---|
| vanilla SASRec | 0.0841 | 0.1054 | 0.0622 | 0.0691 |
| SASRec+latent reasoning | 0.0681 | 0.0880 | 0.0506 | 0.0570 |
| SASRec+latent reasoning+RL (LatentR$^3$) | 0.0801 | 0.0975 | 0.0635 | 0.0691 |
| Ours | 0.1137 | 0.1327 | 0.0915 | 0.0977 |

Table 14: The results of applying LatentR$^3$ to enhance AlphaRec.

| Datasets | Cds | | | | Toys | | | |
|---|---|---|---|---|---|---|---|---|
| **Metrics** | **N@5** | **N@10** | **H@5** | **H@10** | **N@5** | **N@10** | **H@5** | **H@10** |
| AlphaRec | 0.0278 | 0.0373 | 0.0479 | 0.0774 | 0.0347 | 0.0448 | 0.0579 | 0.0893 |
| AlphaRec-latent | 0.0373 | 0.0482 | 0.0629 | 0.0963 | 0.0352 | 0.0465 | 0.0577 | 0.0932 |

**Latent reasoning in LLM.** Latent reasoning has been proposed to address two major limitations of explicit CoT reasoning: (1) high inference latency and (2) the excessive generation of non-essential tokens. Several efforts have been made to explore latent reasoning mechanisms. For example, Deng et al. (2023) introduced the implicit CoT by encoding reasoning directly into the model's internal hidden states, while Hao et al. (2024) proposed treating these hidden states as thoughts in the input space to represent reasoning. However, both methods rely on supervision from explicit CoT data. Goyal et al. (2024) proposed using learnable <pause> tokens to represent reasoning steps, but this approach may suffer from low expressivity (Hao et al., 2024). Among the existing works, the method proposed in (Geiping et al., 2025) is most related to ours. It introduces a recurrent architecture for generating deep latent reasoning, but it is not designed using reinforcement learning and requires pertaining a LLM model with new architecture, making it incompatible with direct integration into existing LLMs.

## O   THE USE OF LARGE LANGUAGE MODELS

The use of Large Language Models (LLMs) is an integral part of our methodology. We employ LLMs primarily to assist in the refinement of textual content, including the optimization of language and clarity in the writing of this paper. A detailed description of how the LLM was utilized within our methodological framework is provided in Section 3. It is important to clarify that while LLMs were used as tools to support drafting and editing, all research ideas, conceptual design, data analysis, and final decision-making were conducted entirely by the human authors.

