# OpenReview forum: "Reinforced Latent Reasoning for LLM-based Recommendation"
_ICLR.cc/2026/Conference — ICLR 2026 Poster_

### Official Review · Reviewer_UkVV · 2025-10-25

**Soundness:** 2
**Presentation:** 3
**Contribution:** 3
**Rating:** 6
**Confidence:** 4

**Summary:**

This paper introduces Reinforced Latent Reasoning for Recommendation ($\text{LatentR}^3$), an end-to-end framework that enables Large Language Models (LLMs) to perform implicit reasoning for recommendation. The core idea is to replace costly, explicit Chain-of-Thought (CoT) generation with compact latent tokens during inference, thus significantly boosting efficiency and removing the dependency on explicit CoT supervision data. $\text{LatentR}^3$ employs a novel two-stage training scheme that leverages Supervised Fine-Tuning (SFT) followed by Reinforcement Learning (RL) using a modified GRPO algorithm (LR-GRPO). The results show that $\text{LatentR}^3$ achieves state-of-the-art performance on various Amazon datasets while maintaining low inference latency.

**Strengths:**

1. The framework successfully addresses a major practical limitation of LLM-based recommenders by eliminating the generation of verbose explicit CoT text. By compressing the reasoning into a few latent tokens, it achieves high performance while maintaining efficiency, which is crucial for real-time deployment.

2. $\text{LatentR}^3$ utilizes a reinforcement learning approach to optimize the latent reasoning process, which allows the model to learn effective reasoning strategies directly from the recommendation outcome without the need for expensive, subjective, and difficult-to-obtain explicit CoT supervision data.

**Weaknesses:**

1. The experimental validation is restricted in scope. Firstly, it only uses a specific family of datasets (Amazon review data), lacking tests on other popular and structurally different public benchmarks like MovieLens-1M. Secondly, the framework is only implemented and tested on relatively small-scale LLM backbones (e.g., $D^3$ or BIGRec, likely based on BERT or similar models), leaving its scalability and continued effectiveness on large, cutting-edge foundation models (e.g., Llama-7B/13B) unproven.

2. While the LR-GRPO method is technically described, the rationale behind critical design choices, such as why the negative PPL is used as a continuous reward proxy and how the batch-level advantage specifically resolves stability issues in the continuous policy space, could be explained more intuitively for a general audience.

**Questions:**

1. The paper states that performance improvements are mainly derived from better recommendations for long-tail items. Since the LLM's rich semantic representations already contribute to better long-tail performance in many methods, how can the authors isolate and demonstrate that the observed gain is primarily due to the learned reasoning captured by the latent tokens, rather than simply an enhanced utilization of the existing semantic information?

2. In the recommendation scenario, how does the performance of implicit reasoning (LatentR³) directly compare to the performance of explicit CoT? If there is a performance gap, what is its magnitude? Understanding this gap is crucial for determining whether implicit reasoning is merely a faster, but less accurate, alternative.

3. The paper utilizes negative Perplexity (PPL) as a continuous reward. What results would be achieved if the model were trained using a simpler binary reward (direct ranking reward or a binary reward signal) or another form of reward, and how would this impact the training stability and final recommendation performance?

4. The compared methods are all purely based on LLM representation and are not combined with traditional ID-based recommendation. Can this approach truly outperform methods where LLM representations are used as input features for traditional recommenders? In this context, can implicit reasoning lead to better LLM representations?

6. Furthermore, is this implicit reasoning scheme applicable to non-textual input, Item Indexing-based recommendation methods, such as 'Adapting Large Language Models by Integrating Collaborative Semantics for Recommendation'?

---

> ### Author Response · Authors · 2025-11-22
>
> Dear reviewer,
>
> Thanks for your valuable feedback. We have carefully considered all comments and provide our point-by-point responses below.
>
> **Weakness:**
>
> **W1: Dataset scope and LLM backbone.**
>
> R1:  To address the concern regarding dataset scope, we have added a new dataset, Steam, in addition to Amazon. The results are summarized in the table below, and our method continues to achieve substantial performance gains.
>
> |  | H5 | H10 | N5 | N10 |
> | --- | --- | --- | --- | --- |
> | Caser | 0.0407 | 0.0685 | 0.0262 | 0.0351 |
> | GRU4Rec | 0.0422 | 0.0721 | 0.0271 | 0.0367 |
> | SASRec | 0.0390 | 0.0633 | 0.0251 | 0.0329 |
> | BIGRec | 0.0559 | 0.0821 | 0.0405 | 0.0489 |
> | LatentR3 | 0.0630 | 0.0959 | 0.0451 | 0.0557 |
>
> Regarding the LLM backbone, BIGRec is implemented on Qwen2.5-1.5B, which is a causal language model rather than a BERT-based method. For larger LLMs, we have also experimented with Qwen2.5-3B, and the results (reported in Table 7 of the Appendix) show that our method still performs better. Due to resource constraints, we are unable to conduct experiments with even larger models.
>
> **W2: Design about LR-GRPO.**
>
> R2: For the PPL reward, the goal is to ensure efficiency. We input the target item into the LLM as the prediction and compute its PPL. Each token in the prediction can be evaluated in parallel, which avoids the time-consuming autoregressive generation process.
>
> For the batch-level advantage computation, the goal is to ensure performance. In recommendation, an entire group might be all low-quality samples. However, when using group-level advantage, the formula $A = \frac{r-mean(r)}{std(r)}$ can erroneously assign high advantage scores to these low-quality samples, which is an undesirable outcome. Batch-level methods can help mitigate this problem.
>
> **Question:**
>
> **Q1: How to isolate the performance gain as being due to the learned reasoning of latent tokens, rather than merely from better utilization of the LLM's inherent semantic knowledge?**
>
> A1: To address this issue, we conducted a new experiment in which the reasoning tokens were masked. The results show that masking the reasoning tokens leads to a substantial performance drop, confirming the importance of the reasoning component.
>
> | | N@5 | N@10 | H@5 | H@10 |
> |---|---|---|---|---|
> | BIGRec | 0.0616 | 0.0672 | 0.0757 | 0.0929 |
> | LatentR3 | 0.0754 | 0.0826 | 0.0934 | 0.1160 |
> | + mask token | 0.0644 | 0.0707 | 0.0833 | 0.1031 |
>
> **Q2: About the performance of explicit CoT.**
>
> A2: The results are shown in Table 5 (COT-RL) in our Paper. Our latent reasoning approach outperforms CoT reasoning, providing improvements not only in efficiency but also in accuracy. Regarding the relatively poor performance of CoT, we note that CoT reasoning can be difficult to learn effectively in our experimental setting.
>
> **Q3: Other forms of reward**
>
> A3: We present a comparison with the relevant method (binary reward) in Section 4.3.4. As shown in Section 4.3.4, our method achieves significantly higher training efficiency than the approach; in terms of performance, our method is slightly better in the corresponding dataset, but is generally comparable.
>
> **Q4: Can implicit reasoning lead to better LLM representations?**
>
> A4: To verify this, we conducted additional experiments on AlphaRec, replacing the standard embeddings with those generated by the latent reasoning model to enhance the ID-based model. The results presented in the table below demonstrate that the latent reasoning approach also exhibits considerable value in this scenario.
>
> CDs Dataset
>
> |  | N5 | N10 | H5 | H10 |
> | --- | --- | --- | --- | --- |
> | AlphaRec | 0.0278 | 0.0373 | 0.0479  | 0.0774 |
> | AlphaRec-latent | 0.0373 | 0.0482 | 0.0629 | 0.0963 |
>
> Toys Dataset
>
> |  |  N5 | N10 | H5 | H10 |
> | --- | --- | --- | --- | --- |
> | AlphaRec | 0.0347 | 0.0448 | 0.0579  | 0.0893 |
> | AlphaRec-latent | 0.0352 | 0.0465 | 0.0577 | 0.0932 |
>
> **Q5: Is this implicit reasoning scheme applicable to non-textual input, Item Indexing-based recommendation methods？**
>
> A5: Due to the substantial differences in pipeline and framework, it is difficult to quickly apply our method to “Adapting Large Language Models by Integrating Collaborative Semantics for Recommendation.” However, we have applied our method to SASRec. The results indicate that our method does not always lead to performance improvements. While our RL-based approach can improve performance, initially integrating latent reasoning into ID-based methods may cause performance drops, limiting overall gains. This is likely because enabling latent reasoning in ID-based architectures requires additional design considerations to address their certain challenges.
>
> |  | H5 | H10 | N5 | N10 |
> | --- | --- | --- | --- | --- |
> | vanilla SASRec | 0.0841 | 0.1054 | 0.0622 | 0.0691 |
> | SASRec+latent reasoning | 0.0681 | 0.0880 | 0.0506 | 0.057 |
> | SASRec+latent reasoning+RL | 0.0801 | 0.0975 | 0.0635 | 0.0691 |
> | Ours | 0.1137 | 0.1327 | 0.0915 | 0.0977 |

---

> ### Author Response · Authors · 2025-11-26
> **Looking Forward to Your Feedback**
>
> Dear Reviewer UkVV,
>
> Thank you for your insightful review. We have added further experiments and analyses addressing all your points. If you have any additional suggestions or questions, please feel free to let us know.
>
> We appreciate your time and consideration.
>
> Best regards, The Authors

---

### Official Review · Reviewer_NXRr · 2025-10-31

**Soundness:** 3
**Presentation:** 2
**Contribution:** 3
**Rating:** 4
**Confidence:** 5

**Summary:**

This paper introduces a new framework for applying reasoning capabilities of Large Language Models (LLMs) to recommendation systems. Instead of relying on explicit chain-of-thought (CoT) reasoning — which requires costly CoT data and causes high inference latency — the authors propose a latent reasoning paradigm.

LatentR³ enables the model to reason within a compact latent space rather than generating textual reasoning traces. The framework integrates a new architectural module, LatentRATT, which adds an attention layer on top of the LLM’s decoding layer to produce latent reasoning tokens aligned with the input embedding space.

**Strengths:**

The paper makes a wise and well-justified design choice. For the sequential recommendation problem, latent reasoning is indeed a more suitable approach. Rather than performing explicit chain-of-thought reasoning, which assumes some human-like logical process, the task here is essentially about learning an approximator that fits the pattern of the next purchased (or interacted) item based on historical data.

In fact, most sequential recommendation problems do not lend themselves to explicit CoT reasoning, because there is rarely a clear, interpretable logic that connects a user’s past behaviors to their next purchase. It is challenging to logically deduce what users will buy next just from previous purchase records. This intuition is also reflected in the results, i.e.,  the numbers in the experimental tables are generally low.

**Weaknesses:**

1. In the experiments, the authors use Qwen2.5-1.5B as the base LLM and keep it frozen during training, only updating the LatentRATT module. This raises several concerns.
First, given that Qwen2.5-1.5B is a relatively small model and easy to fine-tune, it would be reasonable to jointly train the entire model rather than freezing the backbone. Such joint optimization might lead to better results.
Moreover, an additional baseline should be included, a fully fine-tuned Qwen2.5-1.5B model trained with standard supervised fine-tuning (SFT) without CoT reasoning. As discussed in the strengths, sequential recommendation essentially involves learning an approximator from historical behavior to future patterns, so a fully fine-tuned LLM without CoT might achieve similar effectiveness. Input is the same as yours, and output groundtruth is the item.

2. According to Table 3, the datasets used in the experiments contain relatively small item pool size. In real-world recommendation systems, the number of items is usually much larger. It remains unclear how well the proposed method would perform on large-scale datasets with hundreds of thousands or millions of items. The scalability and generalization ability in such realistic settings are therefore uncertain.

3. The paper does not discuss how the model handles cold-start scenarios, such as new or unseen items entering the system. Since this is a crucial challenge in recommendation tasks, it would strengthen the paper to provide either a discussion or an experiment addressing it.

4. The Related Work section overlooks several recent papers, such as Rec-R1, which focuses on explicit CoT reasoning for recommendation. The authors should revisit this section and ensure that all relevant recent studies are properly cited and discussed.

> Lin, J., Wang, T., & Qian, K. (2025). Rec-R1: Bridging Generative Large Language Models and User-Centric Recommendation Systems via Reinforcement Learning. Transactions on Machine Learning Research. https://openreview.net/forum?id=YBRU9MV2vE

5. The relative improvement percentages in Figure 2 may not be a fair representation. For unpopular items where the original performance values are already small, even minor absolute gains can appear as large relative improvements. Figure 6 actually shows that the absolute improvements are nearly the same across categories, so using relative percentages can be misleading. It would be better to present both absolute and relative results for a fairer comparison.

6. In the ablation study section, for the variant “w/o LatentRATT”, the paper does not clearly specify the training configuration. It remains ambiguous whether the entire LLM (Qwen2.5-1.5B) is frozen or partially updated during training. Since LatentRATT is removed, clarification is needed about which parameters are optimized in this setting.

7. Inaccurate or vague expressions
* Line 72 (“training RL from scratch”). This phrase is vague. The authors’ intended meaning appears to be “RL without SFT warm-up”, but the current wording could be misread as training the LLM from random initialization using RL. A clearer expression like “RL-only” would avoid this ambiguity.
* Line 79 (“GRPO corresponds to binary reward”) — The statement is not entirely accurate. In general, policy gradient methods can operate on arbitrary scalar rewards. The current phrasing incorrectly suggests GRPO is inherently binary by definition.
* Line 682 (“as shown in Appendix ???”) — There is an apparent LaTeX reference error (a missing \ref target), which should be corrected for completeness and clarity.

**Questions:**

See the weaknesses.

---

> ### Author Response · Authors · 2025-11-22
>
> Dear reviewer,
>
> Thanks for your insightful comments. We have carefully considered your comments and would like to respond to your concerns as follows.
>
> **Weakness**
>
> **W1: Lack of full fine-tuning of the LLM backbone**
>
> R1: Sorry for the misunderstanding. In our experiments, the Qwen2.5-1.5B backbone is fully fine-tuned for baselines such as BIGRec and D3. For our method, “freezing” Qwen2.5-1.5B also refers only to the stage in which we update the reasoning components, not the entire training pipeline. Thus, all compared methods include performing full backbone fine-tuning in our experiments in the paper. As shown in the paper results, our approach achieves substantially better performance.
>
> **W2: Item pool size.**
>
> R2: We have included a new dataset, Steam, which contains more items (15K). The results are summarized in the table below, and our method continues to deliver substantial performance improvements. Notably, many recent ID-based methods are also evaluated on datasets with item counts similar to those in our experiments, e.g., [1,2]. As shown in the table, our method could still achieve huge performance improvements.
>
> [1]  Yupeng Hou et.al. ActionPiece: Contextually Tokenizing Action Sequences for
> Generative Recommendation. ICML 25.
>
> [2] Yupeng Hou et.al., Generating Long Semantic IDs in Parallel for Recommendation. KDD 2025.
>
> |  | H5 | H10 | N5 | N10 |
> | --- | --- | --- | --- | --- |
> | Caser | 0.0407 | 0.0685 | 0.0262 | 0.0351 |
> | GRU4Rec | 0.0422 | 0.0721 | 0.0271 | 0.0367 |
> | SASRec | 0.0390 | 0.0633 | 0.0251 | 0.0329 |
> | BIGRec | 0.0559 | 0.0821 | 0.0405 | 0.0489 |
> | LatentR3 | 0.0630 | 0.0959 | 0.0451 | 0.0557 |
>
> **W3: Cold-start scenarios.**
>
> R3: Many recommendation methods suffer from cold-start issues, especially traditional ID-based approaches. Since LLM-based methods naturally leverage textual information, they can better handle cold-start items once the LLM is aware of their existence. Our method, which incorporates reasoning, has the potential to deal with new items even more effectively.
>
> For unseen items, we conducted an experiment comparing our method with BIGRec on fully new items. Our method achieves better performance. We report the H@10 results in the following table.
>
> |  | Toys | CDs |
> | --- | --- | --- |
> | BIGRec | 0 | 0 |
> | LatentR3 | 0.0029 | 0.0263 |
>
> **W4: Related work about Rec-R1.**
>
> R4: We will include Rec-R1 in the paper. Based on its latest ArXiv version, Rec-R1 focuses on optimizing LLM-driven text generation—such as query rewriting and summarization—and then uses the generated text as input to downstream recommendation models. In Rec-R1, the recommendation module can be independent of the LLM, and reinforcement learning is applied to optimize the LLM’s text generation (e.g., query rewriting).
>
> In contrast, our work follows the LLM-as-recommender paradigm. The LLM directly predicts the next item, and our design requires the model to first produce its reasoning and then output the final recommended items, aligning more closely with reasoning-centric LLMs such as DeepSeek-R1. In our framework, both reasoning and recommendation are essential components of the LLM’s generation process.
>
> **W5: Why do we use the relative improvement percentages in Fig. 2.**
>
> A5: Since long-tail items naturally have a lower performance upper bound, evaluating them using absolute values may also not be entirely fair. To address this, we report relative improvements using SASRec as a reference, highlighting the effectiveness of our method on long-tail items.
>
> As shown in the table below (where the absolute results N@10 are also included), for the Toys dataset, BIGRec achieves comparable relative improvements over SASRec for long-tail and popular items (17.4% vs. 13.7%). In contrast, LatentR3 shows a significantly larger relative improvement for long-tail items compared to popular items (53% vs. 27.5%). This indicates that while BIGRec’s gains over SASRec are similar across item popularity levels, LatentR3 benefits much more on long-tail items.
>
> Combined with the results from cold-start scenarios, these findings provide strong evidence of the effectiveness of our approach on the long-tail segment.
>
> | Toys | Pupular | RI | Long-tail| RI |
> | --- | --- | --- | --- | --- |
> | SASRec | 0.0812    | - | 0.0315 |  |
> | BIGRec | 0.0923 | **13.7%** | 0.0369 | **17.14%** |
> | LatentR3 | 0.1035 | **27.5%** | 0.0479 | **52.1%** |
>
> **W6: Training configuration about the variant “w/o LatentRATT”**
>
> R6: In this setting, the Qwen2.5-1.5B is fully optimized. We will clarify this setting in the paper revision.

---

> ### Author Response · Authors · 2025-11-26
> **Looking Forward to Your Feedback**
>
> Dear Reviewer NXRr,
>
> Thank you for your insightful review. We have added further experiments and analyses addressing all your points. If you have any additional suggestions or questions, please feel free to let us know.
>
> We appreciate your time and consideration.
>
> Best regards, The Authors

---

### Official Review · Reviewer_SRNz · 2025-11-01

**Soundness:** 2
**Presentation:** 4
**Contribution:** 3
**Rating:** 4
**Confidence:** 4

**Summary:**

This paper introduces an SFT-then-RL paradigm to enable latent reasoning in LLM-based recommendation. The main contributions are:
1. An additional attention layer, LatentRATT, that produces latent vectors and improves performance.
2. An improved reward design that computes the advantage by averaging across the mini-batch rather than across multiple rollouts from the same prompt.

Two further reward modifications are proposed, though the experiments do not fully convince me of their effectiveness. Experiments are conducted on four public datasets. Code is available during review.

**Strengths:**

1. Timely exploration of latent reasoning for recommendation.
2. The LatentRATT layer appears effective for generating latent vectors that improve performance.
3. The advantage computation using batch-level averaging is a clear and reasonable change to GRPO.
4. Extensive experiments on four public datasets.
5. Code is available during the review period.

**Weaknesses:**

1. Why are LLMs necessary here?
    * The authors state that prior latent-reasoning-for-recommendation works are not LLM-based and that their approach is tailored to LLMs.
    * However, the core techniques, namely LatentRATT and the modified GRPO algorithm, are not inherently tied to language modeling. In principle they could be applied to conventional recommenders such as SASRec. The "reasoning" is encoded in a latent vector of length 1, not in explicit linguistic reasoning.
    * To support the claim that the approach is specifically tailored to LLMs, further discussions and an ablation replacing the LLM backbone with a standard recommender would be important.
2. Limited verification of the GRPO-style modifications.
    * My understanding is that batch-level advantage aims to improve performance, while the sampling change and the PPL-based rewards target efficiency. For the two modifications aimed at efficiency, the paper reports only an overall comparison against vanilla GRPO, without isolating the contribution of each change.
    * If sampling and PPL rewards also affect recommendation performance, not only on efficiency, there should be ablations on ranking metrics. If they primarily affect efficiency, there should be ablations on cost and latency. Neither is shown.
3. Insufficient discussion of concurrent latent-reasoning work. The paper briefly mentions two concurrent papers on latent reasoning for recommendation, but does not clearly compare similarities and differences. A more systematic discussion would help clarify novelty, especially given the question of whether LLMs are needed.
4. Data processing choices may bias results.
    * The datasets are not large, and the authors further subsample to about 5k items. This setting resembles a sparse, cold-start-like scenario that may not reflect production traffic and may favor LLM-based methods relative to conventional baselines, weakening the overall contribution.
    * The paper claims to follow established processing methodology but does not cite explicit references.
5. Metric inconsistency in the main text. Figure 2 reports H@10 and N@10, while Figure 3 reports H@5 and N@5. Although full results appear in the appendix, the inconsistency in the main text can feel like cherry-picking.
6. Related work on latent reasoning in LLMs is thin. Only three papers are cited. A more comprehensive and organized survey of latent reasoning for LLMs would be better.
7. Typos and formatting.
    * Line 388 appears to have an unintended newline.
    * Line 682 has a missing section number ("Appendix ??").
    * Line 689 uses \citet{} where \citep{} seems intended.

**Questions:**

Please first refer to the "Weaknesses". Two other questions are:

1. Equation (8): Why does the KL divergence become zero? Isn't LatentRATT a part of the policy network $\pi_{\theta}$?
2. Why does scaling the base model from 1.5B to 3B result in much worse performance, as suggested by the cross-comparison between Table 1 and Table 7?

---

> ### Author Response · Authors · 2025-11-22
> **Response to Reviewer SRNz (Part 1)**
>
> Dear reviewer,
>
> Thank you for the time and effort you dedicated to reviewing our paper. We appreciate the opportunity to clarify and address the concerns you raised, and we hope that our responses satisfactorily resolve them.
>
> **Weakness:**
>
> **W1: Replacing the LLM backbone with a standard recommender to show that LLMs are necessary.**
>
> R1：As stated in the Introduction, this work aims to enhance reasoning in LLM-based recommendation without relying on explicit reasoning data, so our focus remains on LLMs. All components of our method are developed under the LLM-based recommendation setting. Although the method can be forcibly applied to traditional recommenders, its effectiveness there is uncertain due to the natural differences between LLMs and traditional recommenders, for example, their inherent ability gap in various aspects.
>
> ---
>
> Following your suggestion, we applied our method to SASRec, and the results are reported in the tables below. As shown, SASRec does not exhibit meaningful gains after incorporating our method. These results align with the findings from ReaRec’s ablation studies —  applying LLMs’ latent reasoning into traditional methods without specifically solving problems like reasoning degradation does not lead to effective improvements.
>
> |  | H5 | H10 | N5 | N10 |
> | --- | --- | --- | --- | --- |
> | vanilla SASRec | 0.0841 | 0.1054 | 0.0622 | 0.0691 |
> | SASRec+latent reasoning | 0.0681 | 0.0880 | 0.0506 | 0.057 |
> | SASRec+latent reasoning+RL | 0.0801 | 0.0975 | 0.0635 | 0.0691 |
> | Ours | 0.1137 | 0.1327 | 0.0915 | 0.0977 |
>
> **W2: The design motivations for the batch-level advantage, the sampling change, and the PPL-based rewards, along with their corresponding ablation evaluations.**
>
> R2: The batch-level design aims to improve performance, while the PPL-based rewards aim to enhance efficiency. The change in sampling arises from the fact that we can no longer follow the sampling procedure used in the GRPO paper. Reparameterization-based sampling is a straightforward choice for handling latent reasoning. Even when implementing the original GRPO under latent reasoning, we still need to adopt our sampling method.
>
> Regarding the ablations, we already have the corresponding ablations:
>
> |  | Performance (H@10) | time |
> | --- | --- | --- |
> | +sampling+PPL rewards (i.e., w/o Batch Advantage) | 0.1002 | 76 mins |
> | +sampling + PPL rewards+ batch-level advantage (i.e., LatentR3) | 0.1160 | 81 mins |
>
> As shown, the performance improvement is mainly attributed to the batch-level advantage, and this batch-level advantage does not contribute to efficiency improvements. Please note that, in terms of time efficiency, our method is more efficient — it takes only about one quarter of the runtime of the original GRPO (under the changed sampling), as reported in Figure 4 of our paper.
>
> **W3: More detailed discussion of two concurrent papers: ReaRec and STREAM-Rec.**
>
> R3: First, the concurrent papers are designed for traditional recommenders. The main similarity is that they also use latent representations to express latent reasoning. The key differences are as follows:
>
> 1. **ReaRec** does not focus on reinforcement learning to support better exploration of reasoning.
> 2. **STREAM-Rec** focuses more on building a recommendation foundation model with reasoning abilities from scratch. It involves pretraining, SFT, DPO, and RL. In contrast, our work focuses on stimulating LLM reasoning abilities for recommendations. Technically, there are major differences as well: a) STREAM-Rec still relies on pseudo labels to train slow-thinking abilities; b) Their RL stage largely follows the original GRPO design, except that their reward computation incorporates recommendation-specific elements. Our sampling and advantage computation are different.
>
> Regarding whether LLMs are needed, in response to your Comment 1, we have provided experiments showing that they are necessary. Beyond your comments, we also believe LLMs are indeed needed for recommendation, as supported by a growing body of influential work in this direction, e.g., [1,2].
>
> [1] Bowen Zheng et.al. **Adapting Large Language Models by Integrating Collaborative Semantics for Recommendation. 2024. Citation 250+**
>
> [2] Yupeng Hou et.al. **Large Language Models are Zero-Shot Rankers for Recommender Systems. 2024. Citation 550+**

---

> ### Author Response · Authors · 2025-11-22
> **Response to Reviewer SRNz (Part 2)**
>
> **W4: Item pool size and pre-processing method.**
>
> R4: We have included a new dataset, Steam, which contains more items (15K). The results are summarized in the table below, and our method continues to deliver substantial performance improvements. Notably, many recent ID-based methods are also evaluated on datasets with item counts similar to those in our experiments, e.g., [1,2]. As the results show, our method could still achieve huge performance improvements.
>
> Regarding our data pre-processing, the details are provided in the Appendix, and we follow the method [3] to deal with the data.
>
> [1]  Yupeng Hou et.al. ActionPiece: Contextually Tokenizing Action Sequences for
> Generative Recommendation. ICML 25.
>
> [2] Yupeng Hou et.al., Generating Long Semantic IDs in Parallel for Recommendation. KDD 2025.
>
> [3] Keqin  Bao et.al. Decoding Matters: Addressing Amplification Bias and Homogeneity Issue
> for LLM-based Recommendation. EMNLP 24 main.
>
> |  | H5 | H10 | N5 | N10 |
> | --- | --- | --- | --- | --- |
> | Caser | 0.0407 | 0.0685 | 0.0262 | 0.0351 |
> | GRU4Rec | 0.0422 | 0.0721 | 0.0271 | 0.0367 |
> | SASRec | 0.0390 | 0.0633 | 0.0251 | 0.0329 |
> | BIGRec | 0.0559 | 0.0821 | 0.0405 | 0.0489 |
> | LatentR3 | 0.0630 | 0.0959 | 0.0451 | 0.0557 |
>
> **Question:**
>
> **Q1: Why does the KL divergence become zero?**
>
> A1: Since our RL training only fine-tunes the LatentRATT module, which is solely involved in sampling reasoning steps and does not affect answer generation (after we get the reasoning steps), when we feed the sampled results back into the original reference model, the LLM itself remains unchanged. As a result, the logits for the answers remain identical, ultimately leading to a KL divergence of zero.
>
> **Q2：Why does scaling the base model from 1.5B to 3B result in much worse performance, as suggested by the cross-comparison between Table 1 and Table 7?**
>
> A2: The results reported in Table 1 correspond to full tuning, whereas those in Table 7 correspond to LoRA tuning.

---

> ### Author Response · Authors · 2025-11-26
> **Looking Forward to Your Feedback**
>
> Dear Reviewer SRNz,
>
> Thank you for your insightful review. We have added further experiments and analyses addressing all your points. If you have any additional suggestions or questions, please feel free to let us know.
>
> We appreciate your time and consideration.
>
> Best regards,
> The Authors

---

> ### Comment · Reviewer_SRNz · 2025-11-26
>
> I thank the authors for their time and effort in addressing my concerns.
>
> **Motivation**
>
> I acknowledge that "improving LLM-based recommendation via latent reasoning" is novel in the sense that it is underexplored, and I agree that the proposed method looks promising with reasonable and impressive experimental results.
>
> However, to be frank, the motivations provided, specifically that (1) the work targets LLM-based recommendation and reasoning (Rebuttal R1), and (2) LLM-based recommendation is popular (Rebuttal R3), are not entirely convincing. It feels less like a principled motivation and more like an attempt to combine two trendy topics simply because they are popular.
>
> I believe future readers (myself included) may be confused about what distinguishes LLM backbones from traditional methods (like SASRec) when applying this RL-based latent reasoning enhancement, especially given that the approach does not appear to leverage any inductive bias specific to language modeling.
> * Is the improvement an effect of scaling laws? By using LLM-based backbones, do they possess a scale that makes them uniquely capable of being enhanced by latent reasoning?
> * Or does the benefit stem from a language modeling prior?
>
> **Dataset Scales**
>
> Regarding W4, the authors state: "Notably, many recent ID-based methods are also evaluated on datasets with item counts similar to those in our experiments, e.g., [1,2]." I did check the cited papers and found significant discrepancies:
> * The cited papers use datasets with 12k–64k items and 176k–1M interactions.
> * The datasets used in this paper have 5k–6k items and 60k–94k interactions.
>
> Given this, it is inaccurate to claim these papers operate at a similar scale, as there is roughly a 10x difference in the number of items and interactions.
>
> Nevertheless, the experiments on the new Steam dataset appear solid (even taking into account the 1/10 user subsampling, which was not explicitly disclosed in the rebuttal, but claimed in Bao et al. (2024)).
>
> ------
>
> Overall, the rebuttal addresses most of my other concerns. Accordingly, I will raise my score to a positive rating.

---

> ### Author Response · Authors · 2025-11-28
> **Response and Thanks for Raising the Score**
>
> We sincerely thank the reviewer for the reply and for raising the scores. Here are some our additional responses:
>
> 1. **Reasoning+LLM-based recommendation:**
>
> The motivation is much more than just combining two topics. To further respond to you, we first discuss whether the LLM-based recommendation is meaningful. We believe it is, for two reasons:
>
> a). Traditional recommendation models are built based on a recommendation-specific view of the world. LLMs, by contrast, offer a broader perspective, leveraging richer information that can improve recommendations. Moreover, from a capability standpoint, LLMs are also more powerful than traditional models. Our goal is to utilize these capabilities and the information they provide fully.
>
> b). Beyond enhancing recommendations with LLMs, we also consider whether LLMs themselves need recommendation—or more broadly, personalization—abilities. We believe they do, and ideally, because users may have the corresponding needs when using LLMs.
>
> Building on the need for LLM-based recommendation, the next question is how to enhance it. Reasoning is a natural choice. However, explicit reasoning is challenging for recommendations due to the difficulty of obtaining supervised data and inference latency. This motivates us to explore latent reasoning methods for LLM-based recommendation.
>
>
> 2. **Dataset scales:**
>
> We mean that the new dataset is comparable to those used in the cited papers. It contains 108K users, 15K items (>12K in the references), and 314K interactions (>176K in the references).
>
> Thanks,
> The Authors

---

### Author Response · Authors · 2025-12-03
**To AC: Summary of Reviews and Discussion**

Dear Area Chair,

We sincerely appreciate your meta-review of our paper, especially given the substantial additional workload caused by the OR leaking. For your convenience, we provide a concise summary of the reviews and discussion process.

Our paper initially received scores of 6, 4, and 4. **Before the OR leaking incident, one reviewer had already agreed to raise their score from 4 to 6**, leaving the comment: *“The idea is novel, the method is promising, and the experimental results are reasonable and impressive.”* Below, we summarize the key points per reviewer.

**Reviewer 1 (SRNz):**

The main concerns previously raised were: (1) performance when replacing LLMs with traditional models, (2) ablation studies, and (3) the dataset. Our rebuttal, along with additional experiments, addressed these concerns, and **the reviewer acknowledged our response by raising their rating to 6** before the OR leaking event. Remaining points included:

1. *Dataset item pool:* We provided additional statistics demonstrating comparability with reference papers.
2. *Motivation for latent reasoning in LLM-based recommendation:* We clarify that our approach is not a simple combination; it is motivated by the inherent need for efficient reasoning in LLM-based recommendation.

**Reviewer 2 (NXRr):**

This reviewer’s main concerns were experimental. **There was a misunderstanding regarding the baselines, assuming they were not fully tuned in our experiments; in fact, the baselines were fully tuned already**. Similar to Reviewer 1, concerns about the item pool size were addressed experimentally. Additional points:

1. *Cold-start issues:* We experimentally demonstrated that our method outperforms baselines in cold-start scenarios.
2. *Related work (rec-r1):* We provided a detailed discussion highlighting fundamental differences—rec-r1 focuses on query rewriting and summarization rather than general reasoning.
3. *Relative improvement concerns:* We clarified improvements both in absolute and relative terms.

Unfortunately, this reviewer did not participate in the discussion phase, but we confirm that all their concerns have been addressed, particularly the misunderstandings.

**Reviewer 3 (UkVV):**

Reviewer 3 provided positive scores. Their questions included:

1. *CoT performance, alternative reward forms, and LLM-based backbone:* These points were already included in the paper.
2. *Additional datasets:* We added a larger dataset and demonstrated the effectiveness of our method.

Other questions focused on our method’s ability to enhance LLM representations and apply to ID-based approaches. Our experiments confirmed that while our method strengthens LLM representations, it does not yet generalize well to ID-based methods.

We hope this summary provides a clear overview of the review and discussion process and assists in your final evaluation.

Sincerely,
Authors

---

### Meta-Review · Area_Chair_BHTH · 2025-12-30

**Summary:**

This paper presents a new framework, named Reinforced Latent Reasoning for Recommendation (LatentR^3), which applies the reasoning capabilities of LLMs to recommendation systems. LatentR^3 is an end-to-end training framework that leverages reinforcement learning (RL) to optimize latent reasoning without relying on any chain-of-thought (CoT) data. The major technical contributions include an additional attention layer, LatentRATT, and an improved reward design. Extensive results are reported and discussed.

Reviewers agreed that this paper studies an under-explored problem and presents a novel solution to LLM-based recommendation. The idea is well motivated, and the use of reinforcement learning is meaningful and well justified. Experimental results also demonstrated the effectiveness of the proposed method.

Meanwhile, reviewers raised some concerns technical details, experimental settings, baselines, datasets, related work, etc.

**Reviewer Concerns:**

The authors have provided very detailed responses, including ablation studies and results on a new dataset, which have addressed most of the concerns from reviewers. One remaining concern is the motivation for latent reasoning in LLM-based recommendation. The authors should further clarify it in the final version.

**Reviewer Scores:**

Initially this paper received borderline ratings: 6, 4, and 4. During the discussion period, one reviewer mentioned that they "will raise my score to a positive rating." Thus, I think the overall score of this paper would be positive.

---

### Decision · Program_Chairs · 2026-01-26

Accept (Poster)